# Evolutionary genomics of *Leishmania braziliensis* across the neotropical realm
Senne Heeren [1,2,3] ✉, Mandy Sanders[4,13], Jeffrey Jon Shaw[5], Sinval Pinto Brandão-Filho[6], Mariana Côrtes Boité [7], Lilian Motta Cantanhêde[7,8], Khaled Chourabi [7], Ilse Maes[1], Alejandro Llanos-Cuentas [9], Jorge Arevalo[9], Jorge D. Marco [10], Philippe Lemey [2], James A. Cotton [4,11], Jean-Claude Dujardin [1,3], Elisa Cupolillo [7,8,12] ✉ & Frederik Van den Broeck [1,2,12] ✉

The Neotropical realm, one of the most biodiverse regions on Earth, houses a broad range of zoonoses that pose serious public health threats. Protozoan parasites of the *Leishmania* (*Viannia*) *braziliensis* clade cause zoonotic leishmaniasis in Latin America with clinical symptoms ranging from simple cutaneous to destructive, disfiguring mucosal lesions. We present the first comprehensive genome-wide continental study including 257 cultivated isolates representing most of the geographical distribution of this major human pathogen. The *L. braziliensis* clade is genetically highly heterogeneous, consisting of divergent parasite groups that are associated with different environments and vary greatly in diversity. Apart from several small ecologically isolated groups with little diversity, our sampling identifies two major parasite groups, one associated with the Amazon and the other with the Atlantic Forest biomes. These groups show different recombination histories, as suggested by high levels of heterozygosity and effective population sizes in the Amazonian group in contrast to high levels of linkage and clonality in the Atlantic group. We argue that these differences are linked to strong eco-epidemiological differences between the two regions. In contrast to geographically focused studies, our study provides a broad understanding of the molecular epidemiology of zoonotic parasites circulating in tropical America.

Leishmaniasis is a vector-borne disease that is caused by the protozoan *Leishmania* parasite (Trypanosomatidae) and transmitted by phlebotomine sand flies in tropical regions. It is a spectral disease with many clinical manifestations, including visceral and various forms of cutaneous leishmaniasis (CL)[1]. While visceral leishmaniasis is potentially fatal if the patient is not treated, CL is the most common form of the disease and causes a large burden due to social stigma and humiliation[2]. It is estimated that globally about 700,000 to 1.2 million CL cases occur each year[3]. In South America, the annual CL incidence is estimated to lie between 190,000 and 308,000 cases.

One of the most important causative agents of CL and the most severe mucocutaneous disease in South America is *Leishmania* (*Viannia*) *braziliensis*. This species is part of the *L. braziliensis* clade that consists of multiple divergent subgroups such as *Leishmania peruviana*[4–10]. The clade belongs to the subgenus *Viannia*, a group indigenous to the Americas that encompasses the *Leishmania guyanensis* species complex (including *L. guyanensis*, *Leishmania panamensis* and *Leishmania shawi*), *Leishmania lainsoni*, *Leishmania naiffi*, *Leishmania lindenbergi* and *Leishmania utingensis*. The *Leishmania braziliensis* species is a zoonotic parasite circulating principally

[1]Department of Biomedical Sciences, Institute of Tropical Medicine, Antwerp, Belgium. [2]Department of Microbiology, Immunology and Transplantation, Rega Institute for Medical Research, Katholieke Universiteit Leuven, Leuven, Belgium. [3]Department of Biomedical Sciences, University of Antwerp, Antwerp, Belgium. [4]Welcome Sanger Institute, Hinxton, United Kingdom. [5]Departamento de Parasitologia, Instituto de Ciências Biomédicas, Universidade de São Paulo (USP), São Paulo, Brazil. [6]Department of Immunology, Aggeu Magalhães Institute, Fiocruz, Brazil. [7]Leishmaniasis Research Laboratory, Oswaldo Cruz Institute, Oswaldo Cruz Foundation, Rio de Janeiro, Brazil. [8]Instituto Nacional de Ciência e Tecnologia de Epidemiologia da Amazônia Ocidental, INCT EpiAmO, Porto Velho, Brazil. [9]Instituto de Medicina Tropical Alexander von Humboldt, Universidad Peruana Cayetano Heredia, Lima, Peru. [10]Instituto de Patología Experimental, Universidad Nacional de Salta—Consejo Nacional de Investigaciones Científicas y Técnicas (CONICET), Salta, Argentina. [11]School of Biodiversity, One Health and Comparative Medicine, College of Medical, Veterinary and Life Sciences, University of Glasgow, Glasgow, UK. [12]These authors contributed equally: Elisa Cupolillo, Frederik Van den Broeck. [13]Deceased: Mandy Sanders. ✉e-mail: sheeren@itg.be; elisa.cupolillo@ioc.fiocruz.br; fvandenbroeck@gmail.com

in wild rodents[11,12]. Human infections appear to be a spillover from the sylvatic transmission cycle. In addition, skin lesions due to *L. braziliensis* have also been found in domestic animals such as equines, dogs and cats[13,14], suggesting a peridomestic transmission cycle in some areas. In ecological terms, *L. braziliensis* has typical generalist characteristics that allow it to occupy a broad range of ecological niches. This is highlighted by (i) its high genetic diversity[4–9,15], (ii) its continent-wide distribution, occurring in at least 15 Central and South American countries[8,9,16,17], and (iii) its vast range of different vector[18,19] and reservoir[11] host species. The *L. braziliensis* parasite is thus an ideal model species for understanding the population structure of zoonotic pathogens circulating across the Neotropical realm.

Studies investigating the natural genetic diversity of (members of the) *L. braziliensis* clade based on amplified[4] or restriction fragment length polymorphisms (AFLP; RFLP)[7], multilocus microsatellites[5,20] and whole genome sequence data[6,8,9,21,22] revealed a high genetic heterogeneity partitioned by the environment. At the continental level, there is a clear distinction between *L. braziliensis* populations circulating in the Amazonian and Atlantic rainforests[23]. Parasite molecular heterogeneity appeared to be substantially higher in the Amazon, presumably due to its more diverse vector and reservoir host communities[23]. In Peru and Bolivia, studies have shown that the Amazonian *L. braziliensis* is further subdivided into distinct subpopulations that are associated with specific ecoregions[20,22]. In addition, several genetically divergent ecotypes have been reported across South-America[4–8,10], such as *L. peruviana* that emerged in the Peruvian Andes during forestation changes over the past 150,000 years[8]. These observations highlight the extensive diversity of *L. braziliensis* variants infectious to humans.

Most studies on the natural variation of *L. braziliensis* were restricted in terms of the geographic scope[6–8,20,22,23], limiting our knowledge of the evolution of the parasite across its range. Here, the goal of our study was to map the continental genome variation and population structure of *L. braziliensis* within a broad ecological context. This was achieved by using whole genome sequencing data of 257 cryopreserved parasite isolates sampled in Argentina, Bolivia, Brazil and Peru, covering a wide range of ecological regions including Andean, Amazonian and Atlantic forests. Capitalizing on an unprecedented genome dataset for this major human pathogen, we gain essential knowledge on the molecular epidemiology of CL in South America.

## Results
### *L. braziliensis* consists of genetically divergent ecotypes
Paired-end whole-genome sequence data were generated from promastigote cultures of 188 *Leishmania* isolates and combined with previously generated sequencing data of 69 *Leishmania* isolates, including isolates from different *L. (Viannia)* species for comparative purposes (Supplementary Table 1). The numbering of the distinct *L. braziliensis* groups in our paper (L1, L2, and L3) aligns with the genetically distinct *L. braziliensis* groups described in several key studies: Van der Auwera et al. 2014 (types 1 and 2)[5], Brilhante et al. 2019 (type 1 and type 2)[7], and Van den Broeck et al. 2023 (*L. braziliensis* 1, 2, and 3)[10]. The latter study also introduced a third distinct group (*L. braziliensis* 3), identified in the Pernambuco state of Brazil[6]. We acknowledge that our numbering differs from Odiwuor et al. 2012[4], which referred to the distant *L. braziliensis* L2 as group 3. However, our choice of L1, L2, and L3 reflects the most recent and comprehensive classification in the literature.

The median read coverage was 55× (mean = 56×, SD = 21×, min = 0×, max = 137×). For each genome, we identified intervals (defined as accessible genomic regions) with sufficient read depth (5×), base quality (Phred > 25) and mapping quality (Phred > 25). This led to the exclusion of six genomes either because of low coverage of the accessible regions (N = 3) or due to a combination of low median coverage and a fragmented accessible genome (N = 3) (Fig. 1a). In addition, we excluded seven genomes for downstream analyses due to aberrant allele frequency distributions, which are potentially indicative of mixed infections or contamination (Supplementary Fig. 1). The resulting dataset consisted of a total of 244 high-quality genomes (median = 56×, min = 16×, max = 137×) belonging to the *L. braziliensis* clade

(N = 226), *L. (Viannia) guyanensis* species complex (N = 6; 5 *L. panamensis*, 1 *L. shawi*), *L. (Viannia) lainsoni* (N = 2) and, *L. (Viannia) naiffi* (N = 4). Six genomes showed more complex ancestries and were characterized as interspecific hybrid parasites (Supplementary Results).

Genotyping across the combined accessible genome (25.5 Mb, or 77.7% of the genome) of the 244 genomes disclosed a total of 834,178 bi-allelic single nucleotide polymorphisms (SNPs) called against the reference. Phylogenetic network analyses revealed a similar topological relationship among the major *L. (Viannia)* species as disclosed earlier with reduced marker sets[5,24,25]. *Leishmania lainsoni* was phylogenetically the most distant species to *L. braziliensis* L1 (Fig. 1b), with an average of 290,660 homozygous SNPs called against the *L. braziliensis* M2904 reference, followed by *L. naiffi* (average 205,814 homozygous SNPs), the *L. guyanensis* species complex (average 100,520 homozygous SNPs) and *L. braziliensis* L2 (average 55,709 homozygous SNPs) (Fig. 1b). Two divergent subgroups of the *L. braziliensis* clade, *L. peruviana* (average 298 heterozygous sites) and *L. braziliensis* L3 (average 106 heterozygous sites), were each devoid of heterozygous sites compared to the remainder of the *L. braziliensis* genomes (average 13,601 heterozygous SNPs) (Fig. 1b, c).

Inspection of homozygous and heterozygous SNP counts in our panel of 226 *L. braziliensis* genomes revealed four groups of parasites (Fig. 1b and Table 1), including a large group of *L. braziliensis* parasites found within the Amazonian and Atlantic rainforests (hereafter referred to as L1) (N = 182, including the M2904 reference strain), one group found sporadically in Brazil, Peru and Bolivia that has previously been associated with both human and canine leishmaniasis (hereafter referred to as L2) (N = 4)[4,5,7], one group that has been described solely in the Paudalho municipality (Pernambuco state) in Northeastern Brazil (hereafter referred to as L3) (N = 9)[6], and the well-described *L. peruviana* ecotype that is found within the Peruvian highlands (N = 31)[8]. L2 (60,095 SNPs) showed a significantly larger number of SNPs compared to L1 (30,158 SNPs), L3 (25,620 SNPs) and *L. peruviana* (26,024 SNPs) (pairwise Dunn's tests: Supplementary Table 2). L2 appeared genomically to be the most divergent *L. braziliensis* group, as indicated by its distant position in the phylogenetic network (Fig. 1c) and the high number of homozygous SNPs (55,773 SNPs) called against the *L. braziliensis* reference (Fig. 1b).

In terms of heterozygous SNPs per isolate, L2 (median 4406 SNPs) and in particular L3 (median 113 SNPs) and *L. peruviana* (median 98 SNPs) exhibited a significantly lower number compared to L1 (median 13,766 SNPs) (pairwise Dunn's tests: Supplementary Table 3 and Fig. 1b). This observation is not linked to genomic coverages in these groups: 61× in L2, 51× in L3 and 85× in *L. peruviana*. Additionally, the population allele frequency spectrum of L1 was dominated by low-frequency variants (i.e. 75% of the alleles having a frequency below 0.1), whereas the majority of SNP loci were entirely fixed in L2 (81.42%), L3 (99.1%) and *L. peruviana* (66.66%) (Supplementary Fig. 2). Pairwise Dunn's tests on the pairwise genetic distances (Bray–Curtis dissimilarity) confirmed that L2 (L2–L1: Z = 5.53; p = 6.33e-08), L3 (L3–L1: Z = 14.85; p < 2.2e-16) and *L. peruviana* (Lp-L1: Z = 48.96; p < 2.2e-16) hold a significantly lower genetic variability relative to L1.

A phylogenetic network based on 695,229 genome-wide SNPs highlighted the extensive diversity in L1 where individual genomes were separated by relatively long branches, in contrast to the L3 and *L. peruviana* genomes that appear terminally as single divergent offshoots (Fig. 2a). This was corroborated by PCA: PC1 (29.9%) mainly explained the large diversity in L1, while PC2 (16.6%) and PC3 (10.4%) separated *L. peruviana* and L3, respectively (Fig. 2b, c). Ancestry estimation revealed more insight into the divergence of *L. braziliensis* in South America (Fig. 2d–f) and its association with the environment (Fig. 2g). At the deepest evolutionary level (i.e. K = 2) (Fig. 2d), there was a clear separation between L1 parasites from the Atlantic (i.e. the Eastern Highlands) and Amazonian Forests (i.e. Amazonian–Orinocan Lowlands). *L. peruviana* and L3 appeared as separate parasite groups at K = 3 (Fig. 2e) and K = 4 (Fig. 2f), respectively. Both *L. peruviana* from the Peruvian highlands and L3 from the Pernambuco state in Brazil clustered largely with L1 from the Amazonian rainforests at K = 2,

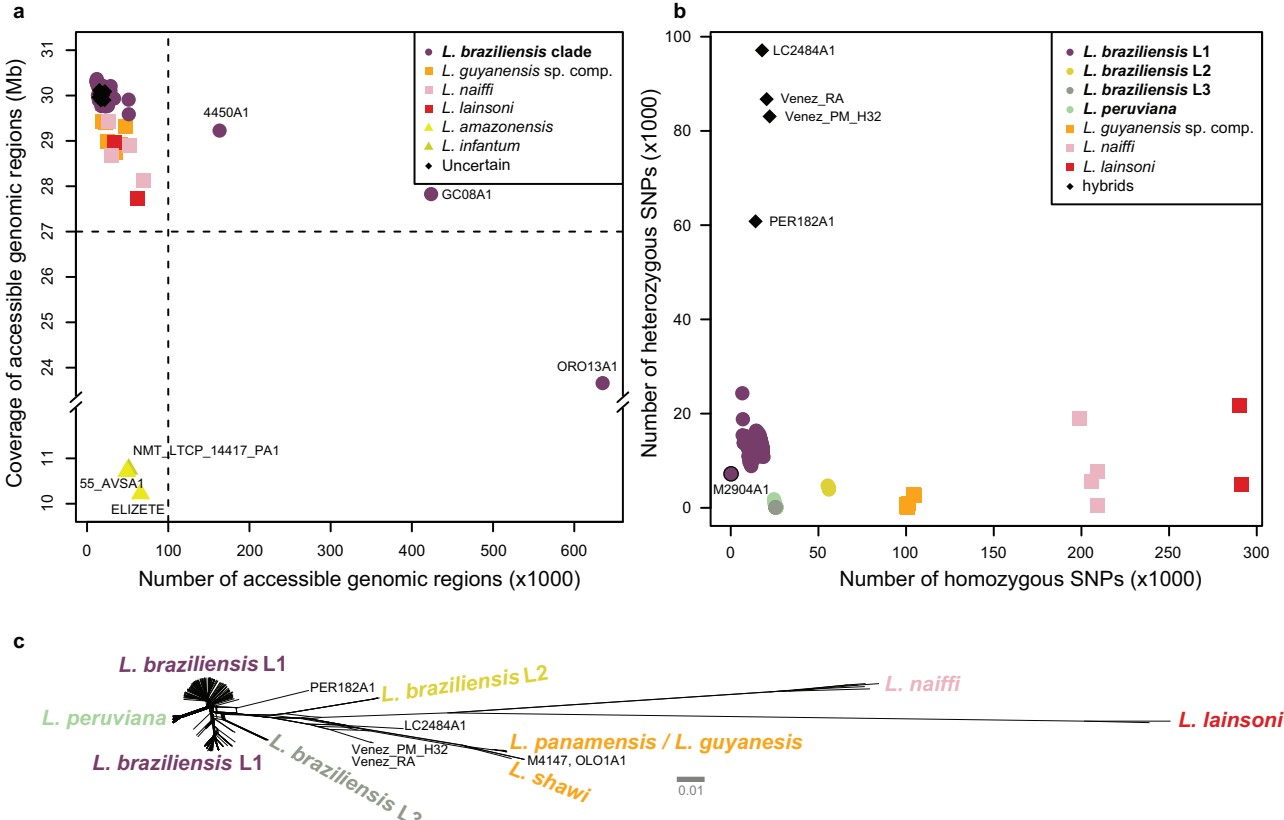

**Fig. 1 | Read coverage and natural genome variation in the *Leishmania* (*Viannia*) subgenus. a** Coverages across the accessible genomes of all 257 isolates. Isolates contained a median of 17.8 k accessible genomic regions, altogether spanning a median of 29.96 Mb (i.e. 91.5% of the haploid genome). Three isolates (55_AVSA1, ELIZETE, and NMT_LTCP_14417_PA1) were removed because of aberrantly low coverage of accessible regions (10.2-10.8 Mb) compared to the other isolates; in silico multi-locus sequencing analysis (MLSA) revealed that these isolates were *Leishmania amazonensis* (55_AVSA1, ELIZETE) and *Leishmania infantum*

(NMT_LTCP_14417_PA1) (results not shown). Three other isolates identified as *L. braziliensis* (4450A1, GC08A1, and ORO13A1) were also removed because of low median coverages (9×–14×) and fragmented callable genomes. **b** Number of homozygous and heterozygous SNPs in the remaining 244 *L.* (*Viannia*) isolates. **c** Phylogenetic network of the 244 *L.* (*Viannia*) isolates based on 834,178 bi-allelic SNPs. Note: the bold legend labels in panels (**a**, **b**) represent the same isolates, all of the *L. braziliensis* clade.

---

although the ancestry of L3 seems somewhat more complex (Fig. 2d). In addition to the clear distinction between Amazonian and Atlantic L1, we also encountered isolates showing patterns of mixed ancestry between these two distinct populations. These isolates originated geographically from the centre of the Amazon, more or less in between the foci of their putative parental lineages.

Pairwise $F_{st}$ calculations confirmed the divergent nature of each parasite group, with estimates ranging from 0.11 to 0.77 (Supplementary Table 4). Notably, $F_{st}$ was similar when estimated between the Amazonian L1 group on the one hand and Atlantic L1 ($F_{st} = 0.12$), L3 ($F_{st} = 0.14$) or *L. peruviana* ($F_{st} = 0.11$) on the other hand, which may indicate that the Amazonian L1 group represents the ancestral parasite population from which all other parasite groups emerged. Estimates of $F_{st}$ were much higher when compared between Atlantic L1 on the one hand and L3 ($F_{st} = 0.39$) or *L. peruviana* ($F_{st} = 0.40$) on the other hand, and between L3 and *L. peruviana* ($F_{st} = 0.77$) (Supplementary Table 4).

**Continental population diversity and structure of *L. braziliensis* L1**
The population structure of the L1 group was examined in more detail based on 194,791 SNPs (178,400 bi-allelic) that were called across 182 *L. braziliensis* isolates sampled in Argentina, Bolivia, Brazil and Peru (Supplementary Table 1). Our analyses predicted that 1727 variants (216 SNPs; 1511 INDELs) have a deleterious impact on the underlying protein sequences. However, the large majority (96.99%) of these deleterious mutations occurred in low frequencies (<5%) (Supplementary Table 5). For

population structure analyses, we retained one genome per clonal group (here-after $N_{unique}$ refers to the number of genomes after removing multiple clones) (see "Methods" and below for more details) and removed SNPs showing high LD, resulting in a dataset of 106,188 bi-allelic SNPs called across 119 genomes.

We identified three major parasite groups showing strong spatio-environmental structuring (Fig. 3a–c), whereby each isolate was assigned with at least 85% ancestry to their respective group. The Atlantic (ATL) group ($N_{all} = 66$, $N_{unique} = 20$) represents parasites isolated in (North-)Eastern Brazil and North Argentina between 1995 and 2016. The West Amazon (WAM) group ($N_{all} = 81$, $N_{unique} = 67$) contained isolates from Bolivia, Western Brazil and Peru that were sampled between 1990 and 2003. The Central Amazon (CAM) group ($N_{all} = 22$, $N_{unique} = 20$) contains isolates sampled in Bolivia and West/Central Brazil between 1984 and 2015 (Fig. 3c). The ATL group (22,414 SNPs) exhibited a significantly lower number of SNPs in comparison to WAM (30,725 SNPs) and CAM (30,254 SNPs) (Kruskal–Wallis test: $\chi^2 = 121.81$; d$f = 2$; $p < 2.2\text{e-}16$; pairwise Dunn's tests: Supplementary Table 6). Isolates showing less than 85% ancestry for any of the inferred groups were grouped together into a conglomerate (CON) group of parasites showing patterns of mixed ancestry ($N_{all} = 13$, $N_{unique} = 12$) (Fig. 3a). Parasites of this polyphyletic group were sampled between 1975 and 2015, originating from Argentina, Brazil and Peru (Fig. 3c).

Next to the geographical east-west stratification of *L. braziliensis* L1, there were also indications of ecological differentiation (Fig. 3b) as we found

**Table 1 | Main characteristics of the identified *L. braziliensis* groups**

| Group | No. isolates | Sampled countries | Sampled ecoregions (level 1)* | Median No. SNPs [min–max] | Median no. heterozygous SNPs | Median no. homozygous SNPs | Refs |
|---|---|---|---|---|---|---|---|
| L1 | 182 | Argentina, Bolivia, Brazil, Peru | Eastern Highland**, Amazonian- Orinocan Lowland, Northern/Central Andes, Gran Chaco | 30,158 [7,385–31,892] | 13,766 | 15,654 | 8,9,22,23 |
| L2 | 4 | Bolivia, Brazil, Peru | Amazonian- Orinocan Lowland | 60,095 [59,879–60,181] | 4406 | 55,773 | 4,5,7 |
| L3 | 9 | Brazil | Eastern highlands | 25,620 [25,596–25,631] | 113 | 25,509 | 6 |
| *L. peruviana* | 31 | Peru | Northern/Central Andes | 26,024 [25,586–26,482] | 98 | 25,664 | 8 |

*Ecoregion classification and data were extracted from data available from the United States Environmental Protection Agency: https://gaftp.epa.gov/EPADataCommons/ORD/Ecoregions/sa/.
**The level 1 ecoregion 'Eastern Highland' encompasses the Atlantic Forest region (a level 2 ecoregion).

a significant association between the three major parasite groups and the biomes where they occur (chi-squared test of independence: $\chi^2 = 300.83$; $df = 18$; $p = 3.24e{-}53$). More specifically, ATL was predominantly linked with the Atlantic Forest biome in Brazil and the Western Dry Chaco in Argentina, CAM was mainly associated with the Amazonian and Coastal Lowlands while WAM was more associated with the Amazonian Irregular Plains and Piedmont, the Yungas, as well as the Central High Andes (Fig. 3b). Pairwise mean $F_{st}$ values revealed a clear differentiation between the Amazonian and Atlantic populations ($F_{st(WAM-ATL)} = 0.16 \pm 0.07$; $F_{st(CAM-ATL)} = 0.15 \pm 0.06$; Supplementary Table 7), which was higher compared to the differentiation within the two Amazonian populations ($F_{st(WAM-CAM)} = 0.06 \pm 0.02$) (Supplementary Table 7). Assuming $K = 5$ populations (as per lowest cross-validation error) revealed the sub-structuring of the WAM population which corresponded with the recently described population structure of Amazonian *L. braziliensis* in Peru and Bolivia[22], and which will not be further discussed here (Fig. 3a, d).

We next investigated the distribution of chromosome and gene copy number variants across the different *L. braziliensis* populations. Consistent with previous reports[26–28], we described considerable variation in chromosome copy numbers, including chromosome 31 that was polysomic in all individuals (Supplementary Fig. 3). A PCA based on gene copy number variations (CNVs) revealed a similar population structure as observed based on SNPs, suggesting that each of the three populations WAM, CAM and ATL carries a specific CNV pattern (Fig. 4a). We found significant differences in the number of CNVs between WAM-CAM and WAM-ATL, though not between CAM-ATL (Fig. 4b, c and Supplementary Table 8). Similarly, the CNV burden (i.e. the proportion of the genome covered by CNVs) ranged between 0.001% and 0.83% of the genome, and was significantly different between amplifications of WAM and ATL, and deletions of WAM and CAM or ATL (Fig. 4d, e; log-rank tests for survival curve differences: Supplementary Table 9). The CNV frequency distributions in each population were skewed towards rare variants, with a median CNV frequency of 5% for WAM, 23% for CAM and 7% for ATL (Supplementary Fig. 4 and Supplementary Tables 10–12). Nine amplifications were present in more than 90% of the individuals in each of the three populations (ANOVA: $F = 268.4$, $df = 2$, $p < 2.2e{-}16$; adjusted $R$-squared = 0.27) (Tukey's HSD test: Supplementary Table 13), seven of which coding for beta-tubulins on chromosome 33 (ORTHOMCL4), one coding for GP63 on chromosome 10 (ORTHOMCL1) and one conserved hypothetical protein on chromosome 31 (ORTHOMCL2303).

**Contrasting recombination histories in Amazonian and Atlantic *L. braziliensis* L1**

We identified 18 clusters of near-identical genomes that constituted 44.5% ($N = 81$) of the isolates (Supplementary Table 1). Genomes within each of the 18 clusters exhibited relatively few heterozygous SNP differences (median = 256, min = 3, max = 2720) and virtually no fixed homozygous SNP differences (homSNPs) (median = 0, min = 0, max = 17) (Supplementary Table 14). These observations suggest that there is a lack of recombination and chromosomal re-assortment between parasites of the same cluster (hereafter referred to as clonal groups). Exceptions were isolates M2903 and EMM (133 homSNPs), LSC358_2 and LSC582 (127 homSNPs), LSC358_2 and LSC684 (127 homSNPs), and LSC358_2 and LSC392 (126 homSNPs). Close inspection revealed that these homSNPs are localized on chromosomes 20 (first 360 kb) and 35 (300 kb–410 kb) for M2903 and EMM or on chromosome 29 (1 Mb–1.2 Mb) for LSC358_2, LSC582, LSC684, and LSC392, and are thus likely the result of gene conversion.

We found a strong difference in the number of near-identical genomes between the Amazonian (WAM and CAM) and Atlantic populations (ATL) (chi-squared test: $\chi^2 = 49.55$; $df = 2$; $p = 1.742e{-}11$; Fig. 5a). In particular, ATL (53/66, 80.3%) appeared to have a significantly higher clonal prevalence compared to WAM (22/81, 27.2%) and CAM (4/22, 18.2%) (pairwise Fisher's exact tests: Supplementary Table 15). No significant differences were found between the populations WAM, CAM and ATL in terms of the

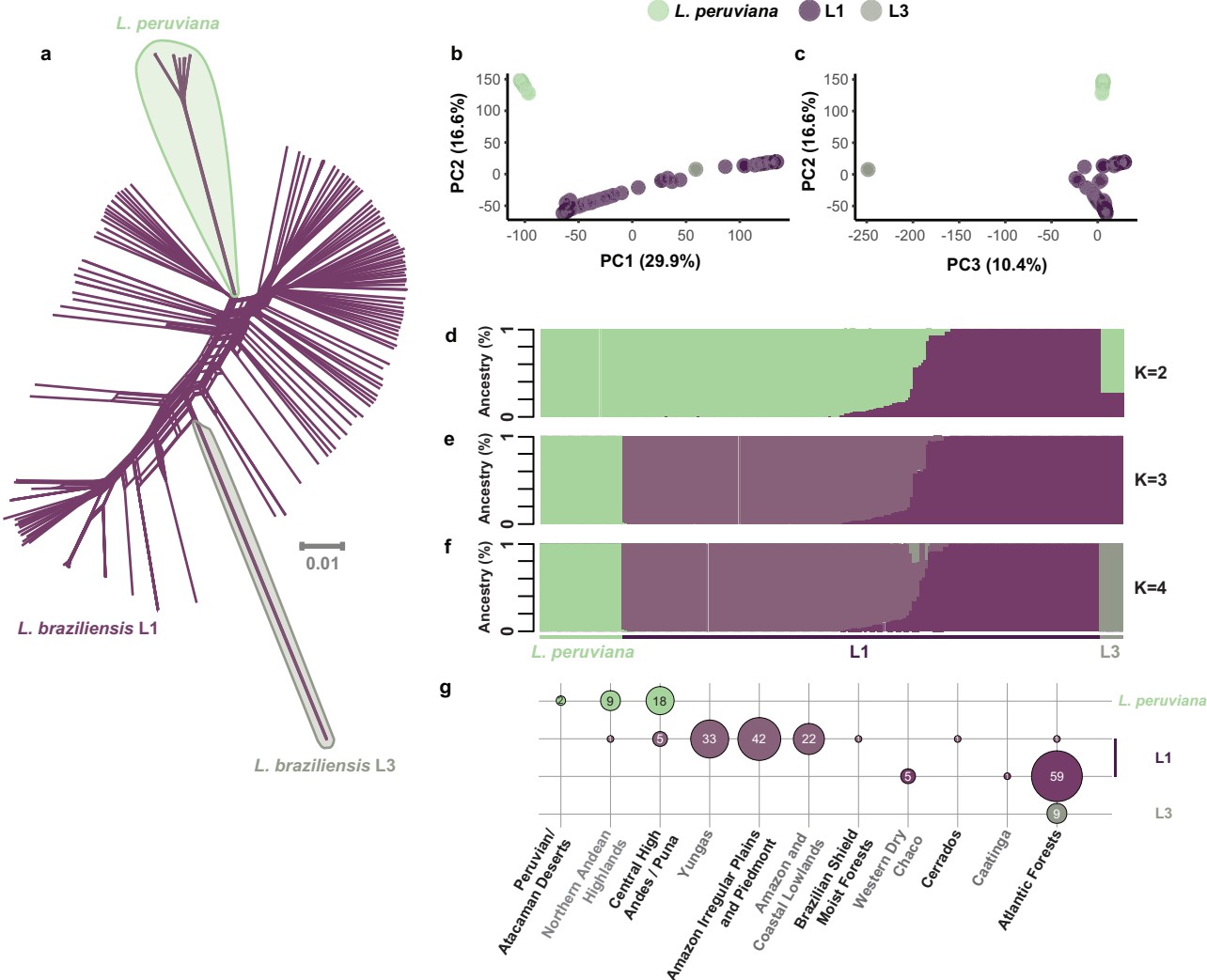

**Fig. 2 | Divergence within the *L. braziliensis* clade. a** A phylogenetic network, based on 695,229 genome-wide SNPs, showing uncorrected *p*-distances between 222 isolates of the *L. braziliensis* clade (incl. L1, L3, and *L. peruviana*). **b, c** Principal component analysis for the 222 isolates showing the first three PC axes. **d–f** ADMIXTURE bar plots showing the estimated ancestry per isolate assuming $K = 2$ (**d**), $K = 3$ (**e**), and $K = 4$ (**f**) ancestral components. **g** Sample size distribution of number of near-identical genomes per clonal group (Kruskal–Wallis test: *Leishmania* isolates from each group and per ecoregion. The four colours match the four ancestral components as inferred with ADMIXTURE $K = 4$ (**f**). Only isolates with at least 70% ancestry for a specific ancestral component were included. Ecoregion data is available from: https://gaftp.epa.gov/EPADataCommons/ORD/Ecoregions/sa/.

number of near-identical genomes per clonal group (Kruskal–Wallis test: $\chi^2 = 7.89$; d$f = 6$; $p = 0.25$). However, three of the top four largest clonal groups (group 2: $N = 32$, group 3: $N = 5$ and group 16: $N = 8$) belonged to ATL, while group 9 ($N = 7$) belonged to WAM. While all but one clonal group (group 8 found in Peru and Bolivia) were unique to a single country, ten groups were additionally restricted to a single department/state (Fig. 5b, c). The remainder of the clonal groups were identified in either two (groups 4, 6, 7, 8, and 17) or three (groups 2 and 16) departments/states.

When only accounting for the unique genomes (i.e. retaining one genome per clonal group) we found that the Amazonian populations (WAM, CAM) were characterized by (i) a strong LD decay ($r^2 < 0.2$ within 10 bp; Fig. 5d and Supplementary Table 16) and (ii) distributions of inbreeding coefficients ($F_{is}$) centred around zero ($0.042 \pm 0.21$ for WAM and $0.009 \pm 0.21$ for CAM) (Supplementary Fig. 5 and Supplementary Table 17). In contrast, ATL showed a much slower LD decay ($r^2 < 0.2$ from 37.1 kb or 101 kb; Fig. 5d and Supplementary Table 16) and distributions of $F_{is}$ deviating negatively from zero ($F_{is} = -0.17 \pm 0.32$) (Supplementary Fig. 5 and Supplementary Table 17). We also observed significant differences between the three populations in the number ($\chi^2 = 33.05$, d$f = 2$, $p = 6.67$e-

08) and proportion ($\chi^2 = 37.33$, d$f = 2$, $p = 7.84$e-09) of 'loss of heterozygosity' (LOH) regions across their genomes (Fig. 5e, f and Supplementary Table 18). Overall, ATL showed a much denser LOH pattern throughout the genome (Supplementary Fig. 6) with an average of 48 LOH blocks, covering an average of 18% of the genome (Supplementary Table 19). In contrast, WAM and CAM each harboured on average 26 and 14 LOH blocks encompassing about 5.7% and 4.2% of the genome, respectively. These results suggest that a considerable degree of genetic diversity has been lost in ATL.

Finally, we inferred the effective population size ($N_e$) of each major *L. braziliensis* L1 population with G-PhoCS (Fig. 6) and simulated the change of $N_e$ over past generations with MSMC2 (Fig. 7). Estimations of $N_e$ were made for different scenarios of historical migration between the Amazon and Atlantic populations: no migration (i), unidirectional migration from the Amazon to the Atlantic (ii), or the Atlantic to the Amazon (iii), and bi-directional migration (iv) (Fig. 6, right panel). All estimations were done in triplicate (i.e. using three different sample subsets) per population and per migration scenario (Supplementary Table 20). This revealed strong significant differences in $N_e$ between all pairwise combinations (main effects

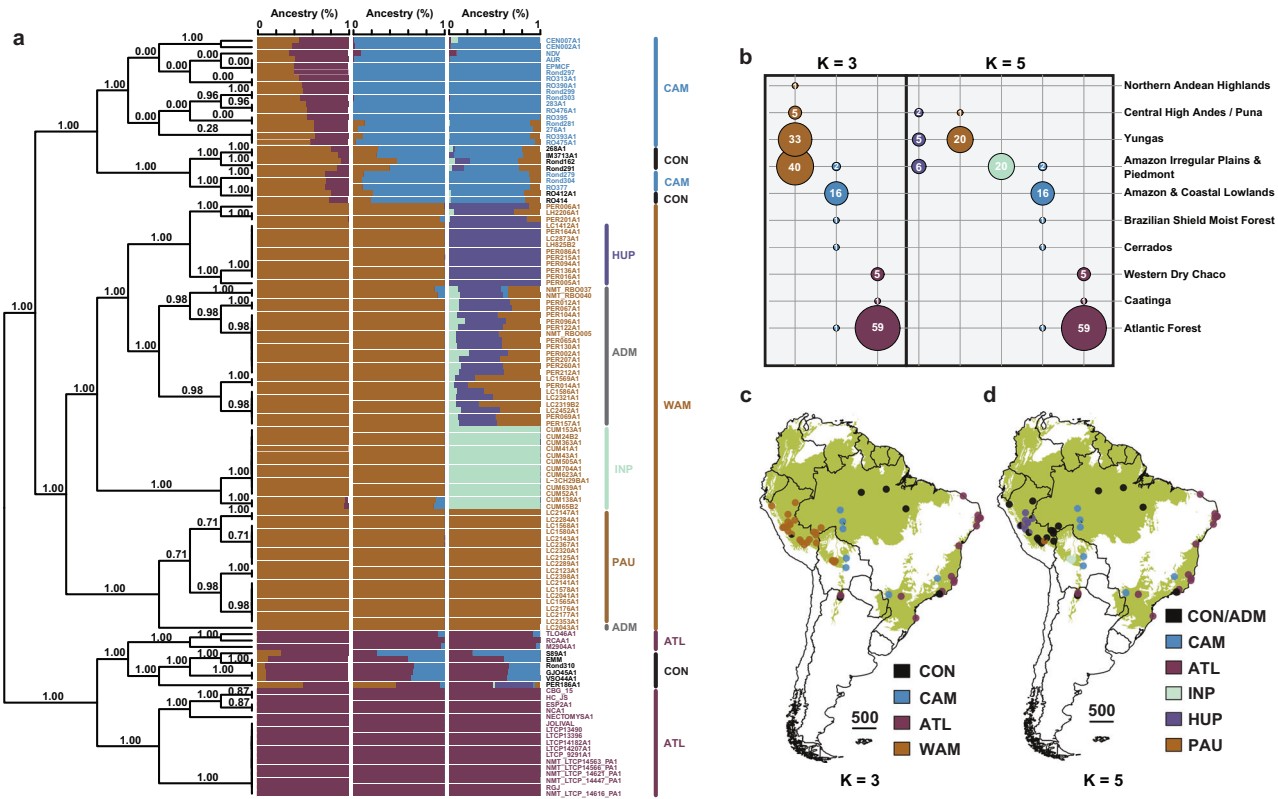

**Fig. 3 | Population genomic structure of *L. braziliensis* L1. a** ADMIXTURE barplots depicting the ancestry per isolate ($N_{unique}$ = 116) assuming $K = 2$, $K = 3$, and $K = 5$ ancestral components. Isolates are labelled according to $K = 3$ ancestral components. Black is used for isolates with uncertain/hybrid ancestry (CON). Outer vertical lines show the major parasite groups (WAM, CAM, ATL, and CON) delineated by ADMIXTURE for $K = 3$. Inner vertical lines represent the parasite groups within WAM as inferred by ADMIXTURE for $K = 5$ in this study, which is in accordance with Heeren et al.[23] (PAU, HUP, INP, and ADM). The left bound tree represents the population tree of L1 as inferred by fineSTRUCTURE. Branch support values represent the posterior probability for each inferred clade. **b** Sample size

distribution per ancestral component per ecoregion (level 2) for all isolates with at least 85% ancestry to a specific group/population. Ecoregion data is available from: https://gaftp.epa.gov/EPADataCommons/ORD/Ecoregions/sa/. **c, d** Map of the South American continent showing the L1 population genomic structure, assuming $K = 3$ (**c**) and $K = 5$ (**d**) populations. The base map depicts the occurrence of (sub-)tropical moist broadleaf forests; data is available from: http://maps.tnc.org/gis_data.html. Country-level data were available from: https://diva-gis.org/data.html. CAM central Amazon, WAM west Amazon, ATL Atlantic, CON conglomerate, PAU Southern Peru, HUP central/northern Peru, INP central Bolivia, ADM admixed.

multi-way ANOVA: $F = 96.55$ on 8 and 423 df; $p < 2.2e-16$; adjusted $R^2 = 0.64$) (Tukey's HSD test: Supplementary Table 21 and Supplementary Fig. 7). Here, ATL consistently exhibited a significantly lower $N_e$ compared to WAM (factor 1.7), CAM (factor 2.6) and AM (i.e. the ancestral Amazonian population prior to the WAM-CAM divergence) (factor 3.1). This pattern remained consistent across the different migration models and replicate runs (Supplementary Table 14 and Supplementary Fig. 7).

Simulations of $N_e$ over time (Fig. 7) revealed similar patterns whereby ATL showed lower $N_e$ compared to WAM and CAM for the past 2.74 million generations. Nevertheless, all parasite populations showed a slight decline in Ne for the past 3 million generations until approximately 400,000 generations ago (Fig. 7). From then on the $N_e$ seemed to rise again for the three populations until 300,000 to 250,000 generations ago when the $N_e$ of WAM and CAM continued to increase whereas ATL exhibited a second and stronger decline. Calculation of the relative cross-coalescence rates (rCCR) between populations revealed mid-point values (i.e. divergence time estimates; see "Methods") at around 500,000–300,000 generations ago for the split between the two Amazonian populations (Supplementary Fig. 8), while this was estimated around 5.2 million to 3.4 million generations ago for the split between the Amazonian and Atlantic populations (Supplementary Fig. 9).

## Discussion

Our study provides the first genome-wide population diversity analysis of the *L. braziliensis* clade at a continent-wide scale. This approach has allowed us to uncover a much finer resolution of the pathogen's evolutionary history, revealing previously undetected patterns of genetic variation and population structure. Our findings significantly advance our understanding of the species' genetic complexity and offer new insights into how environmental factors and anthropogenic disturbances may have shaped parasite population structure across South America.

We confirmed that the *L. braziliensis* clade is genetically highly heterogeneous[4–8,10,23], consisting of divergent parasite groups that are associated with the environment and vary greatly in diversity. We described two major, widespread and genetically diverse groups, one associated with the Amazon and the other with Atlantic Forest biomes, and several smaller groups with little diversity showing a restricted geographic and environmental distribution. Parasites of the smaller groups showed stable long-term genetic diversification and their origin was accompanied by a strong population bottleneck, as indicated by a genome-wide loss of heterozygosity and fixation of SNP polymorphisms. Ancestry and $F_{st}$ estimates suggest that the major admixed Amazonian group may represent the ancestral population from which the other groups emerged, as indicated previously for

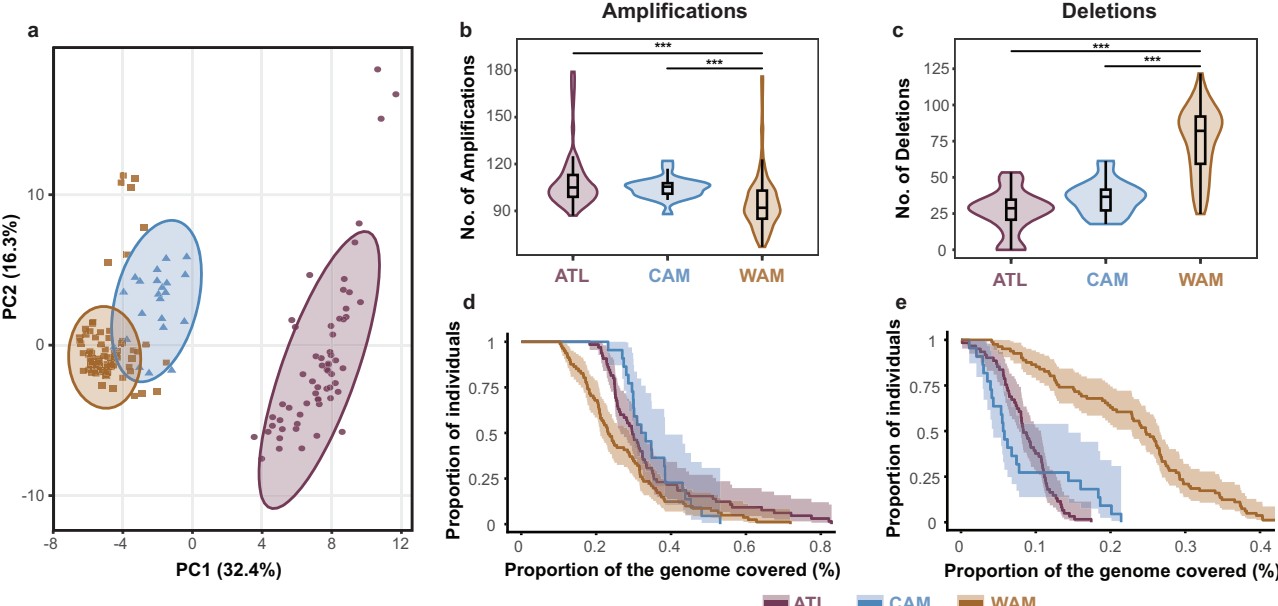

**Fig. 4 | Copy number variations across the major *L. braziliensis* populations.** **a** Scatterplot showing the first two principal components as calculated based on haploid copy numbers of all CNVs, after removing isolates of clonal group 3 (see "Methods"). Ellipses represent the 95% confidence boundaries of the major parasite populations in the PCA space. **b**, **c** Violin plots summarizing the number of CNVs per parasite genome. **d**, **e** Survival curves depicting the CNV burden per *L. braziliensis* population.

*L. peruviana*[8]. This is consistent with historical, biological and epidemiological data suggesting that *L. braziliensis* and its variants preexisted in Amazonia before spreading to other regions through clonal expansion[29]. Our data thus add to a growing body of evidence suggesting the existence of distinct evolutionary and ecological groups of zoonotic *L. braziliensis* parasites in South America[10,16,23].

The main goal was to examine the population diversity and structure of the two major genetically diverse parasite groups that are associated with the Amazon and Atlantic Forest biomes. While both forests were connected as a single forest around 30 thousand years ago (kya)[30], they were separated 20 kya after the last major glaciation[31] by more open savannah-like ecosystems (e.g. Cerrados, Gran Chaco, Caatinga)[32]. These may thus represent important barriers to natural gene flow of *L. braziliensis*[22], as has been suggested for lianas, didelphids and anuran trypanosomes[33–35]. Our demographic models suggest that the two major *L. braziliensis* populations separated 5.2 to 3.4 million generations ago, which would equate to approximately 742 to 340 kya when assuming 7–10 generations per year[36]. The two Amazonian populations diverged much later, namely between 300 and 500 thousand generations ago (30–71 kya). Our results revealed a decline in $N_e$ since about 2.5 million generations ago (357 kya–250 kya) and a strong increase, in particular for the Amazonian populations, about 250 thousand generations ago (35 kya–25 kya). The latter estimate coincides largely with the end of the last major glaciation, which may suggest that subsequent habitat expansions may have promoted a resurgence of this major zoonotic parasite in the Amazon. While these calculations should be considered as rough estimates, they indicate that the history of diversification of *L. braziliensis* is limited to the Pleistocene, an epoch that is characterized by a succession of glacial and interglacial climatic cycles that resulted in habitat fragmentation of *Leishmania*[8].

The two major *L. braziliensis* groups in South America show vastly different recombination histories. The Amazonian group was characterized by high levels of heterozygosity, low linkage disequilibrium and median inbreeding coefficients approximating zero, as would be predicted for a population experiencing predominantly meiotic recombination. In contrast, the Atlantic group was characterized by a high prevalence of near-identical genomes, a slow decay in linkage disequilibrium, negative median inbreeding coefficients and extensive loss of heterozygosity that likely arose from gene conversion events, as would be predicted for a population experiencing predominant clonal propagation. In addition, the effective population size was at least twice as large in the Amazonian groups compared to the Atlantic group. Our results thus clearly show that these protozoan parasites show a broad spectrum of population structures[8,22,37–39]. Within this context, we examined the impact of *L. braziliensis* population structure on the frequency and burden of CNVs, which are characteristic of and highly heterogeneous in *Leishmania*[26,27]. Our data revealed that CNV distributions were strongly skewed towards low-frequency variants in all populations, suggesting that CNVs are deleterious and subject to strong purifying selection in *L. braziliensis*. We hypothesized that CNVs would be more efficiently purged from the large and stable Amazonian parasite populations than from the smaller and endogamous Atlantic populations, as described in the malaria parasite *Plasmodium falciparum*[40]. However, our analysis did not demonstrate that differences in $N_e$ or clonality are sufficient to explain differences in CNV burden and frequency in *L. braziliensis*. This might be because (i) differences in population structures are not strong enough to result in differences in purifying selection, (ii) *Leishmania* is a predominantly diploid organism (in contrast to *P. falciparum* that has a haploid stage), and/or (iii) *Leishmania* can easily change chromosome copy numbers to mitigate the impact of deleterious CNVs[41–43].

We argue that the observed demographic differences may be linked to strong eco-epidemiological differences between the two Forest biomes, in particular differences in the type of transmission cycles[23,44] and forest fragmentation[45]. *L. braziliensis* from the Amazon is predominantly circulating in wild animals where human infections appear as spillovers from the sylvatic life cycle, while *L. braziliensis* from the Atlantic is mainly circulating in animals in both sylvatic and synanthropic foci which may spill over to humans[12,29,46]. Our observation of high parasite diversity in the Amazon compared to the Atlantic Forest is consistent with other studies where sylvatically transmitted parasite populations were more diverse compared to populations predominated by (peri-)domestic transmission[23,44,47,48]. In addition, the Amazon Forest is known as a pristine biome and is the largest contiguous forest in the world. While deforestation in the Amazon poses an extensive threat to the Earth's climate and biodiversity[45,49], the vast majority

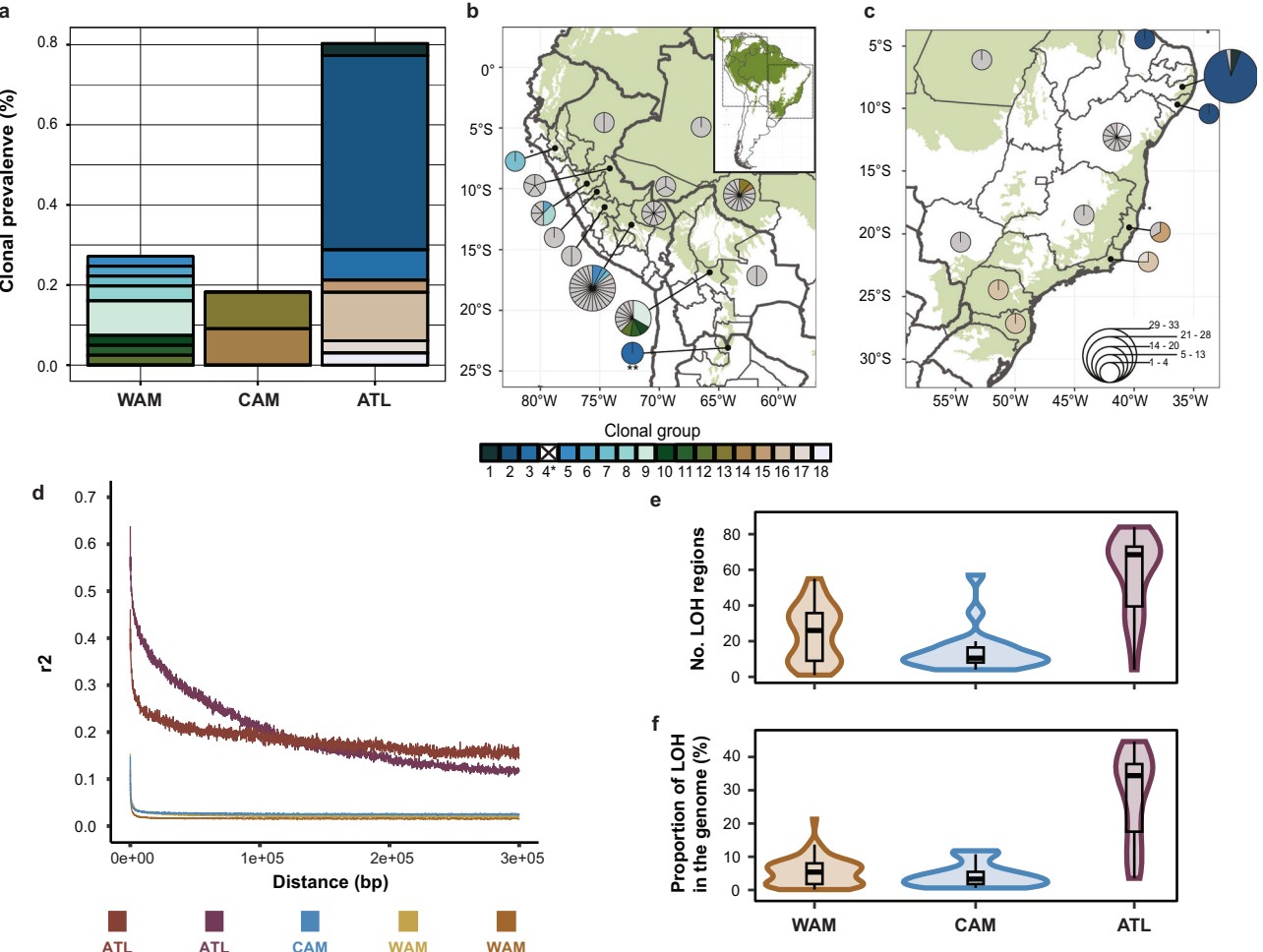

**Fig. 5 | Contrasting clonality and population structure in *L. braziliensis* L1.**
**a** Clonal prevalence per population. **b**, **c** Distribution of genotypes in the Amazon and along the Atlantic coast, summarized per department/state of the respective country. The size of each pie indicates the number of genotypes found in each locality with each segment representing a unique genotype. Coloured segments indicate the different clonal groups that were identified. Note: *clonal group 4 is not included as it consists of two isolates of the CON group; **clonal group 3, located in

Salta, Argentina belongs to ATL. **d** Linkage disequilibrium decay of the different *L. braziliensis* populations, accounting for spatio-temporal Wahlund effects and population size. **e** The number of loss-of-heterozygosity (LOH) regions per major population. **f** Proportion of LOH regions across the genome per major population. For panels (**b**, **c**) the base map depicts the occurrence of (sub-) tropical moist broadleaf forests; data is available from: http://maps.tnc.org/gis_data.html. Country-level data were available from: https://diva-gis.org/data.html.

of the forest remains contiguous[49,50]. In contrast, the Atlantic Forest is known as a degraded biome as it experienced intense deforestation over the past five centuries[51] and is left highly fragmented along the Atlantic coast[45,49,50,52]. Hence, the extensive biodiversity, forest integrity and predominant zoonotic transmission in the Amazon may explain the high diversity of different parasite genotypes sampled in this region, while the genetic uniformity of *L. braziliensis* in the Atlantic may be due to extensive forest fragmentation and predominant synanthropic transmission.

In conclusion, our continent-wide sampling revealed that *L. braziliensis* consists of divergent populations that are associated with the environment and vary greatly in diversity and recombination histories. We argue that these differences are linked to anthropogenic environmental disturbances, such as deforestation and environmental degradation in the Atlantic Forest, that shifted the transmission of *L. braziliensis* from its original sylvatic cycle to a predominantly (peri-) domestic or synanthropic one. These pressures may thus have fuelled clonal expansions of *L. braziliensis* in this region, which may explain the sharp rise in CL cases along the Atlantic coast since the 1980s. *L. braziliensis* thus provides an excellent organism to study a broad spectrum of population structures within a single species, and understand the impact of anthropogenic environmental disturbances on the eco-epidemiology of vector-borne diseases[53].

## Methods

### Parasite culturing and DNA sequencing

This study included 257 isolates from different *Leishmania* (*Viannia*) species, mainly *L.* (*V.*) *braziliensis*, sampled between 1975 and 2016, originating from seven South American countries: Argentina ($N = 11$), Bolivia ($N = 27$), Brazil ($N = 115$), Colombia ($N = 3$), Panama ($N = 2$), Peru ($N = 95$), Venezuela ($N = 2$) and two of unknown origin. Parasite isolates were grown in Schneider culture medium until the end of the log phase at the Oswaldo Cruz Institute (Rio de Janeiro, Brazil. DNA was extracted from $10^7$ to $10^8$ parasites/ml using the QIAmp DNA Mini kit (QIAGEN) following the manufacturer's protocol. Similar to previous work[8], DNA was sheared into 400–600 bp fragments through ultrasonication (Covaris Inc.) and amplification-free Illumina libraries were prepared. One hundred 50 bp paired-end reads were generated on the HiSeq ×10 according to the manufacturer's standard sequencing protocol.

### Variant detection

Paired-end sequencing reads were mapped against the M2904 reference genome, a long-read assembly (available at: https://tritrypdb.org/) comprising the 35 autosomal chromosomes (32.73 Mb) and the complete sequence of the mitochondrial maxicircle (27.69 kb). The mapping of the

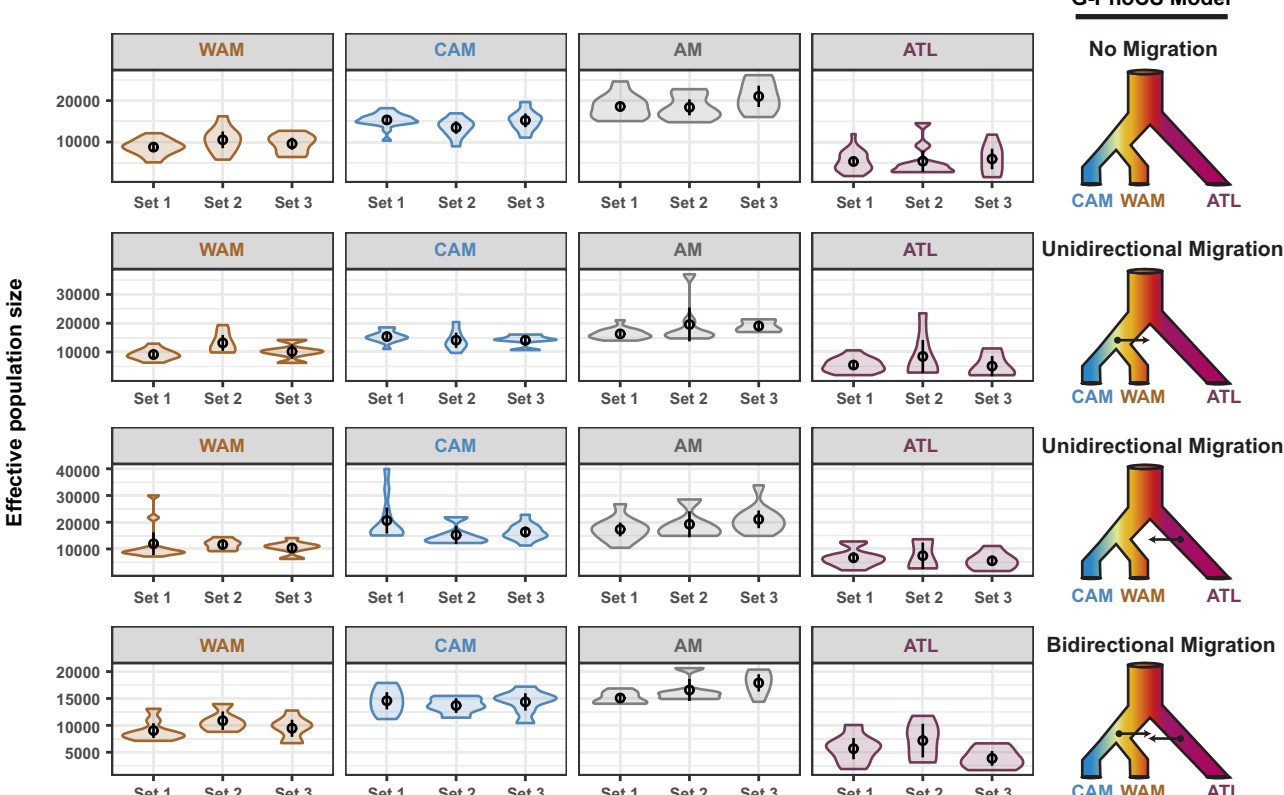

**Fig. 6 | Estimated effective population sizes ($N_e$) per population for four possible migration scenarios.** Each row depicts the $N_e$ estimates per population for a given model of historical migration. WAM, CAM, and ATL represent the three major populations as inferred by ADMIXTURE and fineSTRUCTURE (Fig. 3a). AM represents the ancestral population prior to the split of WAM and CAM. Four models of historical migration were tested: (i) no migration, (ii) unidirectional migration from AM to ATL, (iii) unidirectional migration from ATL to AM, and (iv) bidirectional migration between AM and ATL.

reads was done using SMALT v0.7.4 (available at: https://www.sanger.ac.uk/science/tools/smalt-0). Here we generated the hash index with words of 13 bp long ($k = 13$) that were sampled at every other position in the genome ($s = 2$). Short variants (SNPs and INDELs) were called using GATK's (v.4.0.2) HaplotypeCaller resulting in genotype VCF (gVCF) files for each parasite isolate[54]. Subsequently, all gVCF files were merged using CombineGVCFs after which joint genotyping of all isolates was performed using GenotypeGVCFs. SelectVariants were used to separate SNPs and INDELS which were separately exposed to hard-filtering thresholds using VariantFiltration to exclude low-quality and false-positive variants. SNPs were excluded when: QD < 2.0, FS > 60.0, MQ < 40.0, MQRankSum < −12.5, or ReadPosRankSum < −8.0[55], DP < 5 or when SNPs occurred within SNP clusters (clusterSize = 3 and clusterWindowSize = 10). INDELs were excluded when: QD < 2.0, FS > 200.0, or ReadPosRankSum < −20.0[55]. In addition, we determined which intervals in the genome were accessible for genotyping in each isolate using GATK's CallableLoci (parameters: –minDepth 5 – minBaseQuality 25 – minMappingQuality 25). Finally, we only retained variants that were present in the accessible genome by using the -intersect function of BEDOPS[56].

## Ancestry of *Leishmania* (*Viannia*) species and their hybrids
A phylogenetic network (NeighborNet), based on uncorrected *p*-distances (i.e. the proportion of loci where two sequences differ between each other) of genome-wide, concatenated SNPs, was generated, using the NeighborNet[57] and EqualAngle[58] algorithms implemented in SplitsTree v.4.17.0[59], to infer phylogenetic relationships within the *Leishmania Viannia* subgenus and to identify putative interspecific hybrids (e.g. long terminal branches, reticulated patterns). Hybrid ancestry was subsequently inferred by phylogenetic analysis of (near-) homozygous stretches, as identified by chromosome-specific ARDF (alternate allele read depth frequencies at heterozygous sites)

along with PCA-based hybrid-ancestry estimation using PCAdmix v.1.0[60] using *L. braziliensis* L1, *L. guyanensis/L. panamensism*, and *L. shawi* as putative ancestral groups.

## Ancestry of the *L. braziliensis* clade
A phylogenetic network was constructed in a similar way as described above by calculating pairwise uncorrected *p*-distances based on genome-wide, concatenated, bi-allelic SNPs (683,649 SNPs) using SplitsTree[59]. The ecotype structure was further investigated through i) a principal component analysis (PCA) using the 'glPCA' function of the Adegenet (v.2.1.7) R package[61]; and ii) a simple model-based ancestry estimation, using ADMIXTURE v.1.3.0[62], without prior LD pruning. Differences among the number of SNPs (homozygous or heterozygous) among the different sublineages were tested by means of pairwise Dunn's tests (FSA v.0.9.4 R-package)[63]. A similar comparison was done for comparing the inter-individual pairwise genetic distances, calculated as the Bray–Curtis dissimilarity using the vegan (v.2.6-2) R-package[64], among the different ecotypes. Pairwise $F_{st}$ values between the inferred *L. braziliensis* ecotypes were calculated on a per-site basis over all variable sites using vcftools v.0.1.13 (--weir-fst-pop)[65]. Individual genomes with >70% ancestry for a specific ecotype in the $K = 4$ ADMIXTURE model were included in the $F_{st}$ calculations.

## Identification of near-identical genomes in *L. braziliensis* L1
Similar to a preceding study [22] groups of potential (near-) identical genomes were identified through branch-sharing patterns in the phylogenetic network and low pairwise genetic dissimilarity (<0.02; Bray–Curtis dissimilarity[64]). For each group, fixed SNPs were removed after which counted the number of heterozygous and non-reference homozygous SNPs within each group in a pairwise manner. Near-identical genomes are defined

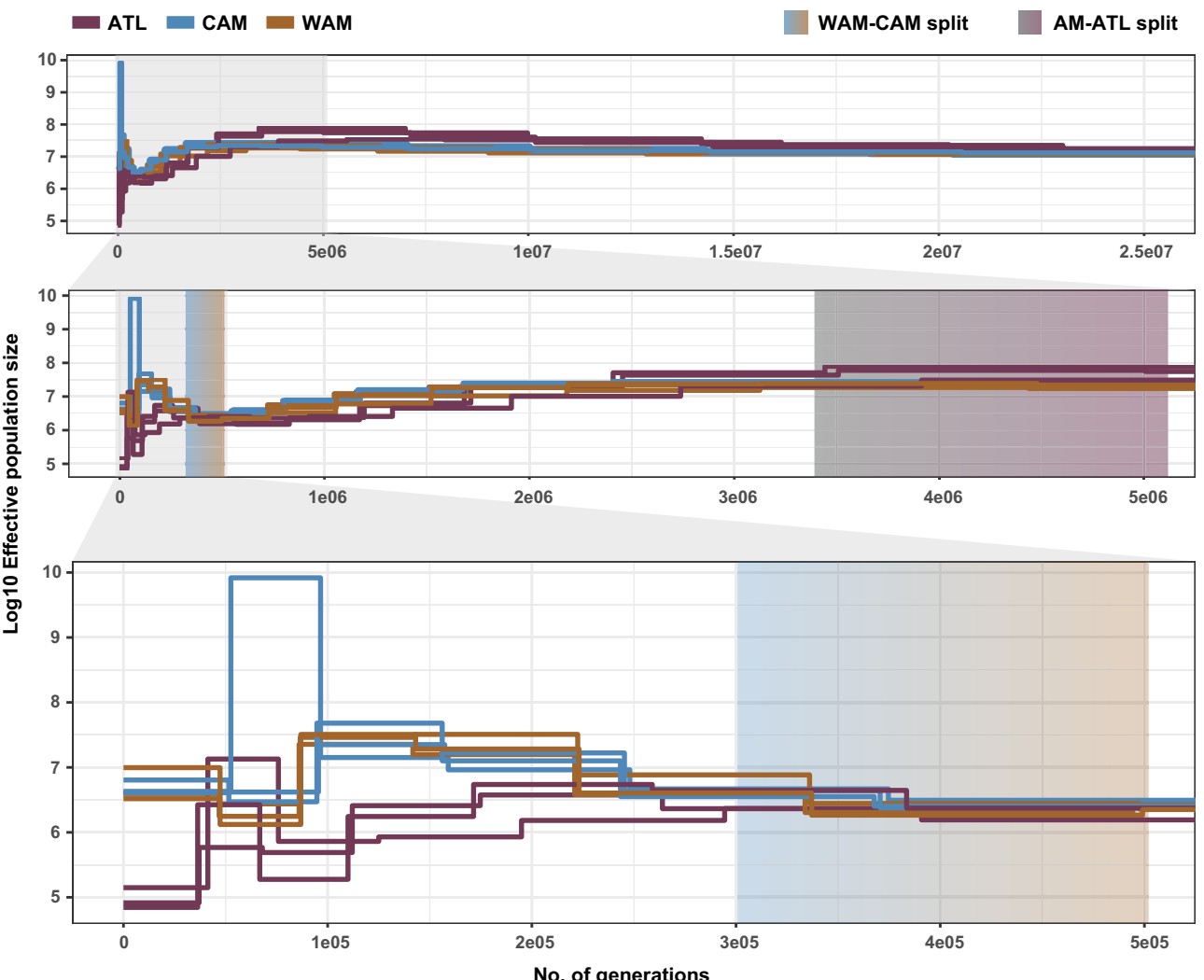

**Fig. 7 | Simulated changes in Ne per population through time (in units of generations ago).** Simulations were performed in triplicate; on the same sample subsets per population as Fig. 6. Gradient boxes depict the estimated time of the first population split (rCCR ≈ 0.5) within the past 25 million generations, between WAM-CAM and AM-ATL based on the relative cross-coalescence rate (Supplementary Figs. 8 and 9). AM = WAM + CAM.

by the (near) absence of homozygous SNPs and a relatively low number of heterozygous SNPs (Supplementary Table 14).

### Population genomic analyses of *L. braziliensis* L1

We constructed a phylogenetic network (NeighborNet) for *L. braziliensis* L1 based on pairwise uncorrected *p*-distances, calculated from genome-wide concatenated SNPs using SplitsTree[59]. The population genomic diversity and structure of *L. braziliensis* L1 were examined in greater detail by two model-based clustering methods: (i) ADMIXTURE v.1.3.0[62] (ii) fineSTRUCTURE v.4.1.1[66]. ADMIXTURE was run on an LD-pruned SNP panel for K = 1–15 populations with a five-fold cross-validation procedure. SNP-pruning was done using plink v.1.9[67] (--indep-pairwise) by retaining SNPs with an $r^2$ lower than 0.3 within 50 bp windows sliding over 10 bp. fineSTRUCTURE was used to infer the genomic ancestries among the individual genomes based on haplotype similarity, generating a co-ancestry matrix. Haplotypes were obtained through computational phasing of the genome-wide SNP genotypes, as was done using BEAGLE v.5.2[68] (default settings). Inferences with fineSTRUCTURE were done after running the algorithm up to 8e06 MCMC iterations (burn-in: 500,000 iterations) and 2e06 max-imization steps (for identifying the best tree-building state). The ecological association with the population structure was tested using a chi-squared test for independence using the CrossTable function from the gmodels

v.2.18.1.1 R package[69]. Pairwise $F_{st}$ estimates between the major parasite groups, as inferred by the K = 3 ADMIXTURE model, were calculated in a similar way as between the different *L. braziliensis* ecotypes. In addition, we also investigated the Hardy–Weinberg equilibrium (HWE) by calculating inbreeding coefficients (Eq. 1); and LD decay was examined using PopLDdecay[70]. To this end, we calculated both $F_{is}$ and LD decay accounting for spatio-temporal Wahlund effects by subsetting individual genomes into groups of individuals close in time (year of isolation < 3 years apart) and space (sample locality in the same department). In addition, the LD decay was corrected for the population sizes (Eq. 2)[71]. The clonal prevalences of the inferred populations were compared by means of a Chi-squared test using the stats R-package[72].

$$F_{is} = 1 - \frac{H_O}{H_E} \qquad (1)$$

($H_O$: observed heterozygosity; $H_E$: expected heterozygosity)

$$r^2_{corrected} = r^2 - \frac{1}{2N} \qquad (2)$$

(*N*: population size)

Finally, we identified loss-of-heterozygosity (LOH) regions across the genome as regions in non-overlapping 10 kb windows[73], for which the following parameters were used[74]: minimum number of SNPs = 1; number of heterozygous SNPs = 0; minimum number of contiguous homozygous 10 kb windows = 4; maximum number of 10 kb gaps allowed within a LOH region = 1/3 of the windows; and maximum number of heterozygous SNPs allowed in a gap region = 2. Differences in the number and proportions of LOH regions among the inferred populations were tested by means of a Kruskal–Wallis test (stats R-package)[72] along with pairwise Dunn's tests with BH corrected *p*-values[63].

### Estimating effective population size (G-PhoCS)

Effective population sizes ($N_e$) were estimated using G-PhoCS v.1.3.2 (Generalized Phylogenetic Coalescent Sampler)[75]. We estimated $N_e$ per chromosome for four different migration models: (i) no migration; (ii) unidirectional migration from the Amazon to the Atlantic; (iii) unidirectional migration from the Atlantic to the Amazon; and (iv) bidirectional migration between the Amazon and Atlantic (Fig. 6). Sequence input files were generated based on VCF and BED files per chromosome after excluding SNPs with a MAF < 0.05 and all SNPs present in CDS regions. The chromosomal VCF and BED files were then converted into the G-PhoCS input format using the 'vcf_to_gphocs.py' script from the Popgen Pipeline Platform (available at: https://github.com/jaredgk/PPP/blob/master/pgpipe/). As G-PhoCS only allows for a small number of individuals per population, we selected three subsets of five isolates per population (WAM, CAM, and ATL) to include in the analyses (Supplementary Table 22). Each G-PhoCS analysis was run over 2,500,000 MCMC iterations (excl. burn-in) sampling every 1000 steps and with an initial burn-in of 500,000 iterations. Additional information on the priors of the G-PhoCS analyses is available in Supplementary Table 22. The following priors were used: tau–theta–alpha = 1; tau–theta–beta = 20,000; mig–rate–alpha = 0.02; mig–rate–beta = 0.0001; locus–mut–rate = CONST; find–finetunes = TRUE; find–finetunes–num–steps = 100; find–finetunes–samples–per–step = 100; tau-initial$_{WAM-CAM}$ = 0.0005; tau-initial$_{AM-ATL}$ = 0.001. Convergence of all theta estimates was assessed by examining their effective sample sizes (ESS) using the Tracerer v.2.2.3 R package[76]. We only included G-PhoCS runs where all theta values reached convergence (i.e. ESS > 200). The posterior distributions of the population size estimates were converted into effective population sizes, using $\theta = 4N_e\mu$, assuming the genome-wide mutation rate ($\mu$) of *Leishmania* spp. is 1.99e-09 per bp per generation[38]. Following Campagna et al. (2015)[77], we limited the interpretations of the $N_e$ estimates to relative differences to rule out potential biases of the assumed mutation rate on the absolute values. Estimates of $N_e$ were compared between parasite populations, as inferred by ADMIXTURE ($K = 3$) and fineSTRUCTURE, by means of a main effect multi-way ANOVA, accounting for the different sample subsets and migration models using the stats R-package[72]. Post hoc pairwise comparisons between populations, migration models and sample subsets were performed using Tukey's HSD (Honest Significant Difference) method (stats R-package[72]).

### Estimating effective population size through time (MSMC2)

Inference of $N_e$ through time was performed using MSMC2 (Multiple Sequentially Markovian Coalescent)[78] and auxiliary scripts from the msmc-tools repository (available at: https://github.com/stschiff/msmc-tools). A mappability mask from the M2904 reference genome was generated using the code from SNPable (available at: http://lh3lh3.users.sourceforge.net/snpable.shtml) and the makeMappabilityMask.py script (msmc-tools). All SNPs were phased using BEAGLE v.5.2[68] and separated per chromosome per individual (only for a subset of individuals; Supplementary Table 22). Chromosome and individual-specific mask files were generated using the vcfAllSiteParser.py script (msmc-tools) after MSMC2 input files were generated using the generate_multihetsep.py script (msmc-tools). Effective population sizes for each *L. braziliensis* population were inferred, in triplicate (Supp Table 23), by running MSMC2 with 500 iterations (-i) and $1 \times 2 + 21 \times 1 + 1 \times 2$ as a time segmentation pattern (-p). The coalescence

rate estimates from MSMC2 were scaled to effective population size values (Eq. 3). The inferred time segments from the MSMC2 output were rescaled to numbers of generations (Eq. 4). Finally, to get an idea of when populations have diverged from each other we calculated the rCCR between WAM and CAM, and between WAM + CAM and ATL. This was achieved by running additional MSMC2 runs for cross-population coalescence rate ($\lambda$) estimates and subsequently running combineCrossCoal.py (msmc-tools) with the msmc2 outputs of the cross-population analysis, as well as the two separate populations as input files. The rCCR was then calculated based on the two within-population coalescence rates and the across-population coalescence rate (Eq. 5). The rCCR ranges between 0 and 1 where a value of 1 points towards the point when both populations probably coalesced into one population while a value of 0 indicates the point when both populations are assumed to be fully separated. The midpoint (rCCR ≈ 0.5) can be seen as an estimate for when both populations have sufficiently diverged from each other to consider them as separate populations[78].

$$N_e = \frac{\frac{1}{\lambda}}{2\mu} \tag{3}$$

($N_e$: effective population size; $\lambda$: coalescence rate; $\mu$: mutation rate)

$$g = \frac{t}{\mu} \tag{4}$$

(*g*: number of generations ago; *t*: time segments; $\mu$: mutation rate)

$$\text{rCCR} = \frac{2\lambda_{POP1-2}}{\lambda_{POP1} + \lambda_{POP2}} \tag{5}$$

(rCCR: relative cross-coalescence rate; $\lambda$: coalescence rate)

### Variant annotation and estimation of chromosome and gene copy number variation in *L. braziliensis* L1

Chromosome and gene copy number variations (CNV) were estimated based on the per-site read depths as obtained with SAMtools depth (-a option)[79]. Chromosomal somy variation was estimated assuming diploidy by multiplying the haploid copy number (HCN) by two. Here the HCN was calculated as the division of the median chromosomal read depth over the genome-wide read depth. Differences in chromosomal copy numbers were tested using the Wilks' lambda test (MANOVA) using the Vegan (v.2.6-2) R-package[64]. In parallel, gene HCNs were calculated by dividing the median read depth per gene, as per coding DNA sequence (CDS), over the genome-wide median read depth. We defined gene CNVs as an increase (*z*-score > 3; amplification) or decrease (*z*-score < −3; deletion) in HCN by subtracting the sample-specific HCN over the genome-wide median HCN. Subsequently, the difference in the number of CNVs and the CNV burden across the genome was assessed between the three L1 populations population by means of a Kruskal–Wallis test (stats R-package[72]) along with pairwise Dunn's tests with BH corrected *p*-values[63] and through survival analyses using the Survival (v.3.3-1)[80] and Survminer (v.0.4.9)[81] R-packages, respectively. The potential difference in HCN of the CNVs that were common in all three populations (i.e. occurring in more than 90% of each population) was tested by means of a one-way ANOVA and subsequent Tukey's HSD post-hoc comparison with *p*-value correction using the BH method. Prior to the ANOVA, several CDS regions were identified as outliers by the Bonferroni outlier test from the car package in R (v.3.1-1)[82]. These CDS regions consistently belonged to isolates of clonal group 3 (Supplementary Table 1), showing extremely high amplifications. For the purpose of comparing the mean HCN per population in these highly frequent amplifications, we discarded these individuals. In addition, a PCA was performed on the HCN of all CNVs using the 'prcomp' function of the stats R-package[72] Variants were annotated, based on the *L. braziliensis* M2904 annotation file as a reference database, using SNPEFF v.5.2[83] with default parameters.

**Reporting summary**

Further information on research design is available in the Nature Portfolio Reporting Summary linked to this article.

## Data availability

Sequence data that were used in this study are available at Sequence Read Archive (SRA) BioProject PRJEB4442, PRJNA1171614, PRJNA521679, PRJNA267749, PRJEB2600, PRJNA484340, PRJEB35158, PRJNA235344, and PRJEB2115. Meta-data on all the *Leishmania* isolates included in this study is provided in Supplementary Data 1. Additional source data are provided in Supplementary Data 2. All other types of data are available from the corresponding authors upon request.

## Code availability

Analyses scripts and input data for the analyses of the effective population size (i.e. G-PhoCS and MSMC2 analyses) are available in Zenodo (https://doi.org/10.5281/zenodo.14050955) and Github (https://github.com/sheerenbiol/Lbra_Ne).

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

## Acknowledgements

This work received financial support from the Directie-Generaal Ontwikkelingssamenwerking en Humanitaire Hulp (DGD) (Belgian cooperation) and was funded in part by the Wellcome Trust grant [206194]. F.V.d.B. and S.H. acknowledge support from the Research Foundation Flanders (grants 1226120N and G092921N). E.C. acknowledges support from Coordenação de Aperfeiçoamento de Pessoal de Nível Superior, Finance Code 001; Conselho Nacional de Desenvolvimento Científico e Tecnológico, Research Fellow, 302622/2017-9; Fundação Carlos Chagas Filho de Amparo à Pesquisa do Estado do Rio de Janeiro, CNE, E26-202.569/2019 and E_ 08/2020 -COLBIO-210.285/2021; GGBN award: GGI-GGBN-2021-278. L.M.C. acknowledges support from Fundação Carlos Chagas Filho de Amparo à Pesquisa do Estado do Rio de Janeiro, Pós-Doutorado Nota 10 E-26/205.730/2022 and 205.731/2022. In Memoriam of Mandy Sanders.

## Author contributions

F.V.d.B., J.-C.D., and E.C. contributed to the conceptualization of the research. Parasite isolation from the samples from Pernambuco was performed by S.P.B.-F. Parasite selection and culturing were performed by I.M., J.D.M., M.C.B., L.M.C., and K.C. Sequencing of the DNA extracts was conducted by M.S. All data analyses were performed and interpreted by S.H. and F.V.d.B. which also stood in the drafting of the manuscript. Manuscript revision was done by S.H., F.V.d.B., E.C., J.C.D., P.L., J.A.C., J.J.S., S.P.B.F., J.D.M., J.A., and A.L.-C.

## Competing interests

The authors declare no competing interests.
