## [Transparent Peer Review file · Communications Biology]

Evolutionary genomics of *Leishmania braziliensis* across the Neotropical Realm

Corresponding Author: Mr Senne Heeren

This manuscript has been previously reviewed at another journal. This document only contains information relating to versions considered at Communications Biology.

Version 0:

Reviewer comments:

Reviewer #1

(Remarks to the Author)

This study provides a genome-wide analysis of the *Leishmania* (Viannia) *braziliensis* complex, a major zoonotic pathogen in Latin America. By analyzing 257 cultivated isolates from across the continent, the research highlights the genetic diversity within this complex and its association with distinct ecological environments. The study proposes the existence of two major groups: one associated with the Amazon biome and the other with the Atlantic Forest, each exhibiting different recombination histories and population structures. The Amazonian group is characterized by high heterozygosity and large effective population sizes, while the Atlantic group shows high linkage and clonality. The authors suggest that these genetic differences are driven by eco-epidemiological factors unique to each region. This approach provides a deeper understanding of the molecular epidemiology of zoonotic parasites in the Neotropical realm, offering insights beyond those provided by more geographically restricted studies.

This work is well-conceived and methodologically sound. The research question is clearly defined, and the analytical tools employed are appropriate and effectively applied to the data. The study offers valuable insights and makes a significant contribution to the field. However, some points need to be revised to enhance the clarity and impact of the findings.

1- The current title does not explicitly mention *Leishmania*, which may lead to a lack of clarity, as "zoonotic parasite" is broad and unspecific. To strengthen the title and better reflect the focus of the study, I recommend revising it to include *Leishmania*. This will make the subject matter more immediately apparent to readers and enhance the impact of the paper.

2- In the Introduction, it is essential to include and clarify the concepts of "complex" and "species." The second paragraph currently refers to *L. v. braziliensis* in general terms without emphasizing that it is a complex encompassing several species. While this may be evident to those working in the field of *Leishmania*, it is essential to state the differences explicitly. It is also needed when *L. lansonii*, *guyanensis*, etc, are mentioned for the first time. This will provide clarity for a broader audience and ensure a more comprehensive understanding of the study's context.

3- In line 267, the sentence "We confirmed that the *L. braziliensis* species complex is genetically highly heterogeneous" lacks clarity, particularly in distinguishing the findings from the extensive body of previous work that has already established this point. It would be helpful to specify how your study adds to the existing knowledge or what unique aspects of genetic heterogeneity were uncovered that were not documented.

4- Why is L2 not shown in Figure 1?

5- The median read coverage reported was 55x with a standard deviation of 21. It is essential to consider whether the depth of sequencing could impact the sensitivity in detecting heterozygosity. The authors mention that L2 (median 4,406 SNPs), L3 (median 113 SNPs), and *L. peruviana* (median 98 SNPs) had significantly lower numbers of heterozygous SNPs per isolate compared to L1 (median 13,766 SNPs). However, L1 represents the largest sample group, which may introduce a potential bias.

It is crucial to discuss this possible bias, as the size of the sample and the sequencing depth can affect heterozygosity

detection.

In this sense, please provide the coverage data for those groups where heterozygosity was reported to be low. If these groups have lower coverage, it could represent an artifact rather than a biological difference. A discussion on this aspect would help clarify whether the observed differences in heterozygosity are genuine or influenced by technical limitations.

Reviewer #2

(Remarks to the Author)

The authors present a thorough and complex evaluation of the population structures of circulating *L. braziliensis* in South America. The data is clear and supports the existence of multiple populations across the species with some geographical boundaries. This study has clearly undergone several rounds of review and editing, and only minor questions remain.

My primary concern is with the possible confusion that the multiple grouping levels might bring: The parasites are first segregated into L1 to L3, then L1 is separated again into ATL, CAM and WAM while L2 and L3 are no longer discussed and later the discussion refers to 'two main groups'.

Regarding the L1 group, based on the analyses presented (in particular Figure 1F) I do believe L1 should be split into the three grouping of ATL/CAM/WAM rather than ATL/AM.

Secondly would it be possible to renumber the grouping to L1-L5 (L1-ATL, L2-CAM, L3-WAM, L4....) or some other consistent grouping?

Figs 1-2: while the study focuses on L1, the identification of divergent L2 and L3 groups is also interesting and I feel L2 should be included in figure 1 since L3 is present and L2 and L3 should be included in panels C and D of figure 2 (or in a new supplementary figure) for thoroughness. As the disease evolves, it is possible these rare lineages will later become emerging foci and their inclusion in this study may be beneficial.

Supp fig1: have the authors explored the possibility of hybrid parasites with aneuploid genomes? For example, panel C for sample LH937B2 may represent a 25%:75% ratio that is seen in tetraploid chromosomes from hybrid parasites. Phasing these SNPs and looking for crossover events may be interesting.

Isolate codes: Isolates codes would need to be reviewed and possibly normalized (to genbank SRA codes?) to make it easier to follow. I also noticed LH937B2 in the figure is listed as LH937A1 in the table, are they different?

Supp fig 13: It seems like the authors chose regions where SNPs matching *L. guyanensis* were dense (top) and regions where SNPs matching *L. shawi* were dense (bottom) to show the samples are related to the two species. The phylogeny presented is therefore limited to a small subset of data that was pre-selected to match one species or the other and could reflect this pre-selection rather than true phylogeny. Other hybrid analyses in the field generally match each half of regions of heterozygous SNPs to homozygous parental SNPs. If a region is dense enough in heterozygous SNPs to be phased by the read length used in this study and each haplotype segregated into separate species this would be ideal.

Version 1:

Reviewer comments:

Reviewer #1

(Remarks to the Author)

The authors considered all the comments and suggestions and answered them all. The manuscript has been substantially improved. I understand that it is now suitable for publication in *Communications Biology*.

One question for the authors: Is it possible that L2 is not *L. braziliensis*? Considering its distance from the rest of the species, it could be a new clade/species not described previously. Just to take it into consideration.

Reviewer #2

(Remarks to the Author)

1. In Reviewer 2 Question1: The numbering is said to be consistent with previous literature. However in [Van der Auwera et al. 2014] the groupings are referred to as *L. braziliensis* type 1 and *L. braziliensis* type 2 and in <https://doi.org/10.1038/s41467-023-44085-2> it is the LRV genotypes that are labeled L1 through L9. Further, in the new Figure 1c, a clear separation of L1 into top and bottom branches is seen. If the current L1-L3 grouping originates from previous literature, it is not clearly stated in the results or introduction and could be clarified [eg. The samples were grouped into the previously established L1-L3 grouping (xxxx et al) for the initial analysis] followed by a breakup in the results when it becomes clear that L1 is composed of multiple groups. As it currently is written, it is unclear why L1-L3 was chosen.
2. Review 1Q4, reviewer 2Q2: The inclusion of the figure depicting the L2 samples is greatly appreciated.
3. Supplementary figures appear to be misnumbered after s9 (they are listed as s2, s3, etc again.) and figures beyond s9 are not referenced in the main text?

4. All other concerns and corrections have been addressed.

Point by point response to the reviewers

Manuscript: Evolutionary genomics of a zoonotic parasite across the Neotropical realm

Reviewer #1:

This study provides a genome-wide analysis of the *Leishmania* (*Viannia*) *braziliensis* complex, a major zoonotic pathogen in Latin America. By analyzing 257 cultivated isolates from across the continent, the research highlights the genetic diversity within this complex and its association with distinct ecological environments. The study proposes the existence of two major groups: one associated with the Amazon biome and the other with the Atlantic Forest, each exhibiting different recombination histories and population structures. The Amazonian group is characterized by high heterozygosity and large effective population sizes, while the Atlantic group shows high linkage and clonality. The authors suggest that these genetic differences are driven by eco-epidemiological factors unique to each region. This approach provides a deeper understanding of the molecular epidemiology of zoonotic parasites in the Neotropical realm, offering insights beyond those provided by more geographically restricted studies.

This work is well-conceived and methodologically sound. The research question is clearly defined, and the analytical tools employed are appropriate and effectively applied to the data. The study offers valuable insights and makes a significant contribution to the field. However, some points need to be revised to enhance the clarity and impact of the findings.

1. The current title does not explicitly mention *Leishmania*, which may lead to a lack of clarity, as "zoonotic parasite" is broad and unspecific. To strengthen the title and better reflect the focus of the study, I recommend revising it to include *Leishmania*. This will make the subject matter more immediately apparent to readers and enhance the impact of the paper.

We agree with the reviewer and changed the title to “Evolutionary genomics of *Leishmania braziliensis* across the Neotropical realm”.

2. In the Introduction, it is essential to include and clarify the concepts of "complex" and "species." The second paragraph currently refers to *L. v. braziliensis* in general terms without emphasizing that it is a complex encompassing several species. While this may be evident to those working in the field of *Leishmania*, it is essential to state the differences explicitly. It is also needed when *L. lansoni*, *guyanensis*, etc, are mentioned for the first time. This will provide clarity for a broader audience and ensure a more comprehensive understanding of the study's context.

We agree with the reviewer and have revised the introduction to clarify the species concepts of *L. braziliensis*. We replaced the term "species complex" with "clade" to more accurately reflect the broader grouping of subgroups currently classified under the *L. braziliensis* clade. The term "species complex" suggests that all subgroups have been formally recognized as different species within *L. braziliensis*, which has not (yet) been fully established.

3. In line 27967, the sentence "We confirmed that the *L. braziliensis* species complex is genetically highly heterogeneous" lacks clarity, particularly in distinguishing the findings from the extensive body of previous work that has already established this point. It would be helpful to specify how your study adds to the existing knowledge or what unique aspects of genetic heterogeneity were uncovered that were not documented.

We appreciate the reviewer's feedback on the clarity of our initial sentence in the discussion. We acknowledge that we may not have clearly distinguished our findings from the existing literature. To address this, we added an introductory paragraph at the beginning of the discussion to emphasize the novel aspects of our research. In the second paragraph we also highlight the findings that have been described in previous studies. The remainder of the text discusses how our work enhances the current understanding of the genetic heterogeneity within the *L. braziliensis* clade.

4. Why is L2 not shown in Figure 1?

We decided to exclude L2 from Figure 2 (previously Figure 1) due to its status as a highly divergent subgroup within the *L. braziliensis* clade. Including L2 could mask the primary focus of the figure by reducing the resolution needed to clearly detect patterns of population structure among *L. braziliensis* L1 and the less divergent subgroups L3 and *L. peruviana*.

However, we understand the importance of providing a broader phylogenetic context to the paper. To address this, we have moved supplementary Figure 2 to the main text as Figure 1. The new Figure 1 includes a phylogenetic network (panel c) of all 244 genomes used for genotyping, clearly illustrating the relationships within the *L. braziliensis* clade (L1, L2, L3, *L. peruviana*) and with other *L. (Viannia)* species.

5. The median read coverage reported was 55x with a standard deviation of 21. It is essential to consider whether the depth of sequencing could impact the sensitivity in detecting heterozygosity. The authors mention that L2 (median 4,406 SNPs), L3 (median 113 SNPs), and *L. peruviana* (median 98 SNPs) had significantly lower numbers of heterozygous SNPs per isolate compared to L1 (median 13,766 SNPs). However, L1 represents the largest sample group, which may introduce a potential bias. It is crucial to discuss this possible bias, as the size of the sample and the sequencing depth can affect heterozygosity detection. In this sense, please provide the coverage data for those groups where heterozygosity was reported to be low. If these groups have lower coverage, it could represent an artifact rather than a biological difference. A discussion on this aspect would help clarify whether the observed differences in heterozygosity are genuine or influenced by technical limitations.

We appreciate the reviewer's concern regarding the potential bias in heterozygous sites due to limited sequencing depth. However, we believe this is not a significant issue in our analysis.

The coverage data requested by the reviewer was included in the initial submission as Supplementary Figure 2a. In response to the reviewer's suggestion, we have now presented this coverage data in the main text as Figure 1. In Figure 1a, it is evident that all but three individuals of the *L. braziliensis* clade, including the low heterozygosity lineages L3 and *L. peruviana*, have similar genome coverages. This finding supports the validity of the observed differences in heterozygosity.

Additionally, we have revised the first two paragraphs of the results section to better address the concerns raised by the reviewer and describe the results visualized in Figure 1 (lines 103-136). These include a sentence emphasizing that genotyping was done across the combined accessible genome, specifically in regions where the read depth was at least 5x in every genome (lines 107-109). In addition, we provided the range of coverages for each of the groups with low heterozygosity levels to demonstrate that these have relatively high coverages (lines 154-155). We believe that these additions to the main text eliminate any ambiguity and demonstrate the validity of our results to the reviewer.

Reviewer #2:

The authors present a thorough and complex evaluation of the population structures of circulating *L. braziliensis* in South America. The data is clear and supports the existence of multiple populations across the species with some geographical boundaries. This study has clearly undergone several rounds of review and editing, and only minor questions remain.

1. My primary concern is with the possible confusion that the multiple grouping levels might bring: The parasites are first segregated into L1 to L3, then L1 is separated again into ATL, CAM and WAM while L2 and L3 are no longer discussed and later the discussion refers to 'two main groups'. Regarding the L1 group, based on the analyses presented (in particular Figure 1F) I do believe L1 should be split into the three grouping of ATL/CAM/WAM rather than ATL/AM. Secondly would it be possible to renumber the grouping to L1-L5 (L1-ATL, L2-CAM, L3-WAM, L4....) or some other consistent grouping?

We appreciate the reviewer's feedback regarding the multiple grouping levels. Our intention was to clearly present the population structure analyses at different evolutionary levels. The first level categorizes the main parasite groups L1, L2, L3 and *L. peruviana*, which are illustrated in Figures 1 and 2. At this level, it is not yet apparent that L1 would be split into the three groups ATL, CAM and WAM, as suggested by the reviewer. This distinction becomes evident at the second level, where more detailed analyses separate the larger L1 group into ATL and AM, with AM further separated into WAM and CAM, as depicted in Figure 3. We hope that this structure clarifies the relationships among the groups.

We would like to maintain the current numbering of the different groups as it is consistent with classifications in previous publications (Van der Auwera et al. 2014; Van den Broeck et al. 2023). We believe that this consistency across studies is crucial to minimize confusion among readers.

2. Figs 1-2: while the study focuses on L1, the identification of divergent L2 and L3 groups is also interesting and I feel L2 should be included in figure 1 since L3 is present and L2 and L3 should be included in panels C and D of figure 2 (or in a new supplementary figure) for thoroughness. As the disease evolves, it is possible these rare lineages will later become emerging foci and their inclusion in this study may be beneficial.

We agree with the reviewer on the importance of providing a broader phylogenetic context in this study. In response to this comment, as well as comments 3 and 4 from Reviewer 1, we have moved Supplementary Figure 2 to the main text as Figure 1. This figure now includes a network of all genomes, clearly depicting the different *L. braziliensis* lineages, including the divergent L2 lineage, in relation to other Leishmania Viannia species.

We decided to exclude L2 from Figure 2 (previously Figure 1) due to its status as a highly divergent subgroup within the *L. braziliensis* clade. Including L2 could mask the primary focus of the figure by reducing the resolution needed to clearly detect patterns of population structure among *L. braziliensis* L1 and the less divergent subgroups L3 and *L. peruviana*.

3. Supp fig1: have the authors explored the possibility of hybrid parasites with aneuploid genomes? For example, panel C for sample LH937B2 may represent a 25%:75% ratio that is seen in tetraploid chromosomes from hybrid parasites. Phasing these SNPs and looking for crossover events may be interesting.

We appreciate the reviewer's suggestion regarding the potential for hybrid parasites with aneuploid genomes. We acknowledge that peculiar read depth distributions observed in certain isolates may point to polyploidy due to hybridization. However, these patterns could also be the result of a mixed infection or contamination.

Isolate LH937 serves as a good example. Initially typed as *L. peruviana*, preliminary genome analyses of the first genomic sequence (labeled as LH937A1; see explanation of nomenclature in our answer to comment 4) revealed a normal alternate allele frequency distribution centered around 0.5 and a clustering entirely with *L. braziliensis*. This inconsistency led us to reculture and sequence the isolate again (labeled as LH937B2), which displayed aberrant allele frequency distributions with a 25%:75% ratio, and a clustering between *L. peruviana* and *L. braziliensis*. While these findings are intriguing, they raise several possibilities, including the likelihood of isolate LH937 being a mixed or contaminated isolate containing both *L. peruviana* and *L. braziliensis*.

However, given that i) the current text and analyses are already quite dense, ii) the main goal of the paper is to elucidate the population genomics of *L. braziliensis* L1 and iii) only a minority of isolates (13 out of 188) exhibited aberrant allele frequencies, we opted not to expand on the specific causes of the anomalous allele frequencies for each isolate presented in Supplementary Figure 1. We plan to address the potential implications of mixed infections in *L. braziliensis* in future

studies, particularly by employing new technologies such as direct genome sequencing with SureSelect.

4. Isolate codes: Isolates codes would need to be reviewed and possibly normalized (to genbank SRA codes?) to make it easier to follow. I also noticed LH937B2 in the figure is listed as LH937A1 in the table, are they different?

We appreciate the reviewer's observation regarding inconsistencies in the isolate codes, and we have revisited the main text and figures to address this issue. The genbank SRA codes were added to Supplementary Table S1 for completion, but we prefer to keep the isolates' codes in figures and text to ensure continuity with previous studies.

The double-digit code appended to most of the isolates' names (e.g. LH937) are standard practice at the Sanger Institute. They typically indicate different genomes of the same isolate whereby "A1" indicates the first genome sequenced, "B2" the second genome, and so forth. We have opted to retain these codes to prevent misunderstandings regarding the versions of the genomes used in the analysis. In our analyses, we used LH937B2 because preliminary analyses of the LH937A1 genome were inconsistent (refer to our response on comment 3). Eventually, we also removed LH937B2 for downstream analyses due to anomalous read depth distributions.

We thank the reviewer for pointing out the error in Supplementary Table 1, which originated from an earlier version of the manuscript. We have corrected LH937A1 to LH937B2 in the table.

5. Supp fig 13: It seems like the authors chose regions where SNPs matching *L. guyanensis* were dense (top) and regions where SNPs matching *L. shawi* were dense (bottom) to show the samples are related to the two species. The phylogeny presented is therefore limited to a small subset of data that was pre-selected to match one species or the other and could reflect this pre-selection rather than true phylogeny. Other hybrid analyses in the field generally match each half of regions of heterozygous SNPs to homozygous parental SNPs. If a region is dense enough in heterozygous SNPs to be phased by the read length used in this study and each haplotype segregated into separate species this would be ideal.

The approach of focusing on homozygous regions is key to identifying the parental genomes. Homozygous stretches represent regions of the genome where one parental species became fixed in the hybrid genome. By focusing on these regions, we are able to pinpoint the parental species with high confidence. In the plots, we examine homozygous regions where one haplotype clearly clusters or matches with the sequence of one of the two parental species (e.g., *L. guyanensis* or *L. shawi*). This method ensures that we identify the parental genomes accurately, even if the hybrid genome contains both heterozygous and homozygous regions. While it is understandable to question whether the (relatively small) regions selected in these phylogenetic analyses reflect true species relationships or are biased by pre-selection, we want to clarify that this part of the analyses represent

an exploration of the potential hybrid nature of these isolates. We substantiated these suspicions by performing a genome wide ancestry estimation which is based on phased SNPs (supplementary figure 14). This gives a broader picture of the hybrid ancestry beyond the specific regions highlighted in the phylogenetic trees of supplementary figure 13.

Point by point response to the reviewers

Manuscript: Evolutionary genomics of a zoonotic parasite across the Neotropical realm

Reviewer #1:

The authors considered all the comments and suggestions and answered them all. The manuscript has been substantially improved. I understand that it is now suitable for publication in Communications Biology.

One question for the authors: Is it possible that L2 is not *L. braziliensis*? Considering its distance from the rest of the species, it could be a new clade/species not described previously. Just to take it into consideration.

We follow the argument of the reviewer. In 2019, a study found that dogs in the Peruvian Andes and in Brazil were infected by *L. braziliensis* type 2 (here referred to as L2), which may thus be considered as the etiological agent of canine leishmaniasis (Brilhante et al, 2019). This means that *L. braziliensis* L2 is both genomically and clinically different from L1, and may thus potentially be considered as a different species. The classification of *Leishmania* (*Viannia*) species is beyond the scope of this paper, but results from our study and others clearly show the need to formally describe the vast diversity of the different *L. braziliensis* lineages, including their epidemiology and clinical significance.

Reviewer #2:

1. In Reviewer 2 Question1: The numbering is said to be consistent with previous literature. However in [Van der Auwera et al. 2014] the groupings are referred to as L *braziliensis* type 1 and L *braziliensis* type 2 and in <https://doi.org/10.1038/s41467-023-44085-2> it is the LRV genotypes that are labeled L1 through L9. Further, in the new Figure 1c, a clear separation of L1 into top and bottom branches is seen. If the current L1-L3 grouping originates from previous literature, it is not clearly stated in the results or introduction and could be clarified [eg. The samples were grouped into the previously established L1-L3 grouping (xxxx et al) for the initial analysis] followed by a breakup in the results when it becomes clear that L1 is composed of multiple groups. As it currently is written, it is unclear why L1-L3 was chosen.

We agree with the reviewer that it is not sufficiently clear from our text why L1-L3 was chosen. Therefore, we included the following paragraph on lines 107-114 at the beginning of the results section for completion:

The numbering of the distinct *L. braziliensis* groups in our paper (L1, L2 and L3) aligns with the genetically distinct *L. braziliensis* groups described in several key studies: Van der Auwera et al. 2014 (type 1 and type 2), Brilhante et al. 2019 (type 1 and type 2) and Van den Broeck et al. 2023 (*L. braziliensis* 1, 2 and 3). The latter study also introduced a third distinct group (*L. braziliensis* 3), identified in the Pernambuco state of Brazil (Figueiredo de Sá et al., 2019). We acknowledge that our numbering differs from Odiwuor et al. (2012), which referred to the

distant *L. braziliensis* L2 as group 3. However, our choice of L1, L2 and L3 reflects the most recent and comprehensive classification in literature.

2. Review 1Q4, reviewer 2Q2: The inclusion of the figure depicting the L2 samples is greatly appreciated.
3. Supplementary figures appear to be misnumbered after s9 (they are listed as s2, s3, etc again.) and figures beyond s9 are not referenced in the main text?

We appreciate the reviewer for highlighting this issue. This was a formatting error where the final PDF was not properly updated with the latest Word version of the main and supplementary text. We have now corrected this, and the numbering of the supplementary figures should be accurate.

Supplementary Figures 10-13 are referenced in the supplementary results section. Additionally, we identified a typo in line 17 of the supplementary text: the reference to Supp. Fig. 9 should be Supp. Fig. 10. This has been corrected.

4. All other concerns and corrections have been addressed.

References:

Brilhante, A. F., Lima, L., Zampieri, R. A., Nunes, V. L. B., Dorval, M. E. C., Malavazi, P. F. N. da S., Melchior, L. A. K., Ishikawa, E. A. Y., Cardoso, C. de O., Floeter-Winter, L. M., Teixeira, M. M. G., & Galati, E. A. B. (2019). Leishmania (Viannia) braziliensis type 2 as probable etiological agent of canine cutaneous leishmaniasis in Brazilian Amazon. *PloS One*, 14(4), e0216291. ([10.1371/journal.pone.0216291](https://doi.org/10.1371/journal.pone.0216291))

Van der Auwera, G., Ravel, C., Verweij, J. J., Bart, A., Schönian, G., & Felger, I. (2014). Evaluation of four single-locus markers for Leishmania species discrimination by sequencing. *Journal of Clinical Microbiology*, 52(4), 1098–1104. ([10.1128/jcm.02936-13](https://doi.org/10.1128/jcm.02936-13))

Van den Broeck, F., Heeren, S., Maes, I., Sanders, M., Cotton, J. A., Cupolillo, E., Alvarez, E., Garcia, L., Tasia, M., Marneffe, A., Dujardin, J.-C., & Van der Auwera, G. (2023). Genome Analysis of Triploid Hybrid Leishmania Parasite from the Neotropics. *Emerging Infectious Diseases*, 29(5), 1076–1078. ([10.3201/eid2905.221456](https://doi.org/10.3201/eid2905.221456))

S L Figueiredo de Sá, B., Rezende, A. M., Melo Neto, O. P. de, Brito, M. E. F. de, & Brandão Filho, S. P. (2019). Identification of divergent Leishmania (Viannia) braziliensis ecotypes derived from a geographically restricted area through whole genome analysis. *PLoS Neglected Tropical Diseases*, 13(6), e0007382. ([10.1371/journal.pntd.0007382](https://doi.org/10.1371/journal.pntd.0007382))

Odiwuor, S., Veland, N., Maes, I., Arévalo, J., Dujardin, J.-C., & Van der Auwera, G. (2012). Evolution of the Leishmania braziliensis species complex from amplified fragment length polymorphisms, and clinical implications. *Infection, Genetics and Evolution: Journal of Molecular Epidemiology and Evolutionary Genetics in Infectious Diseases*, 12(8), 1994–2002. ([10.1016/j.meegid.2012.03.028](https://doi.org/10.1016/j.meegid.2012.03.028))